# Three Different Genetic Risk Scores Based on Fatty Liver Index, Magnetic Resonance Imaging and Lipidomic for a Nutrigenetic Personalized Management of NAFLD: The Fatty Liver in Obesity Study

**DOI:** 10.3390/diagnostics11061083

**Published:** 2021-06-13

**Authors:** Nuria Perez-Diaz-del-Campo, Jose I. Riezu-Boj, Bertha Araceli Marin-Alejandre, J. Ignacio Monreal, Mariana Elorz, José Ignacio Herrero, Alberto Benito-Boillos, Fermín I. Milagro, Josep A. Tur, Itziar Abete, M. Angeles Zulet, J. Alfredo Martinez

**Affiliations:** 1Department of Nutrition, Food Science and Physiology, Faculty of Pharmacy and Nutrition, University of Navarra, 31008 Pamplona, Spain; nperezdiaz@alumni.unav.es (N.P.-D.-d.-C.); bmarin.1@alumni.unav.es (B.A.M.-A.); fmilagro@unav.es (F.I.M.); mazulet@unav.es (M.A.Z.); jalfmtz@unav.es (J.A.M.); 2Centre for Nutrition Research, Faculty of Pharmacy and Nutrition, University of Navarra, 31008 Pamplona, Spain; jiriezu@unav.es; 3Navarra Institute for Health Research (IdiSNA), 31008 Pamplona, Spain; jimonreal@unav.es (J.I.M.); marelorz@unav.es (M.E.); iherrero@unav.es (J.I.H.); albenitob@unav.es (A.B.-B.); 4Clinical Chemistry Department, Clínica Universidad de Navarra, 31008 Pamplona, Spain; 5Department of Radiology, Clínica Universidad de Navarra, 31008 Pamplona, Spain; 6Liver Unit, Clinica Universidad de Navarra, 31008 Pamplona, Spain; 7Centro de Investigación Biomédica en Red de Enfermedades Hepáticas y Digestivas (CIBERehd), 28029 Madrid, Spain; 8Biomedical Research Centre Network in Physiopathology of Obesity and Nutrition (CIBERobn), Instituto de Salud Carlos III, 28029 Madrid, Spain; pep.tur@uib.es; 9Research Group on Community Nutrition and Oxidative Stress, Balearic Islands Institute for Health Research (IDISBA), University of Balearic Islands-IUNICS, 07122 Palma, Spain

**Keywords:** NAFLD, genetic risk score, fatty liver index, lipidomic, magnetic resonance imaging

## Abstract

Non-alcoholic fatty liver disease (NAFLD) affects 25% of the global population. The pathogenesis of NAFLD is complex; available data reveal that genetics and ascribed interactions with environmental factors may play an important role in the development of this morbid condition. The purpose of this investigation was to assess genetic and non-genetic determinants putatively involved in the onset and progression of NAFLD after a 6-month weight loss nutritional treatment. A group of 86 overweight/obese subjects with NAFLD from the Fatty Liver in Obesity (FLiO) study were enrolled and metabolically evaluated at baseline and after 6 months. A pre-designed panel of 95 genetic variants related to obesity and weight loss was applied and analyzed. Three genetic risk scores (GRS) concerning the improvement on hepatic health evaluated by minimally invasive methods such as the fatty liver index (FLI) (GRS_FLI_), lipidomic-OWLiver^®^-test (GRS_OWL_) and magnetic resonance imaging (MRI) (GRS_MRI_), were derived by adding the risk alleles genotypes. Body composition, liver injury-related markers and dietary intake were also monitored. Overall, 23 SNPs were independently associated with the change in FLI, 16 SNPs with OWLiver^®^-test and 8 SNPs with MRI, which were specific for every diagnosis tool. After adjusting for gender, age and other related predictors (insulin resistance, inflammatory biomarkers and dietary intake at baseline) the calculated GRS_FLI_, GRS_OWL_ and GRS_MRI_ were major contributors of the improvement in hepatic status_._ Thus, fitted linear regression models showed a variance of 53% (adj. R^2^ = 0.53) in hepatic functionality (FLI), 16% (adj. R^2^ = 0.16) in lipidomic metabolism (OWLiver^®^-test) and 34% (adj. R^2^ = 0.34) in liver fat content (MRI). These results demonstrate that three different genetic scores can be useful for the personalized management of NAFLD, whose treatment must rely on specific dietary recommendations guided by the measurement of specific genetic biomarkers.

## 1. Introduction

Non-alcoholic fatty liver disease (NAFLD) is the leading cause of liver disease in high-income countries, affecting more than 25% of the population [1]. NAFLD includes a spectrum of liver disease conditions ranging from simple steatosis to non-alcoholic steatohepatitis (NASH) with variable degrees of fibrosis and cirrhosis [2,3], and it has become one of the most common causes of chronic liver diseases [3]. 

The NAFLD etiology is multifactorial and yet incompletely understood, but ultimately appears as determined by the combination of environmental factors, such as excessive adiposity or the presence of type 2 diabetes (T2D) as well as the accumulation of intrahepatic lipids, alterations of energy metabolism, insulin resistance and inflammatory processes, where the genetic make-up may emerge [1,4]. Other factors such as obesity and a sedentary lifestyle, together with metabolic syndrome features and ethnicity, influence the risk of NAFLD [5]. 

The gold standard for the diagnosis of NAFLD is liver biopsy [6]. However, the decision about when to perform this screening remains controversial [2], being necessary the search for less invasive methods for screening patients suspected of this disease [6,7,8,9]. To date, ultrasonography is recommended as the first-line diagnostic method in assessing steatosis [6]. Moreover, magnetic resonance imaging (MRI) has also shown a high accuracy for diagnosing liver fat content [10], as well as advances in the analysis of big data from lipidomic have provided novel insights [11]. Furthermore, non-invasive biomarkers and some validated algorithms such as the fatty liver index (FLI) have also become a useful tool for the diagnosis of simple steatosis and hepatic functionality [12,13]. 

Heritability and family history have a clinically relevant impact on fatty liver disease onset and progression [1,5,14]. In particular, the genetic variants in the genes *PNPLA3* (rs738409), *TM6SF2* (rs58542926), *GCKR* (rs1260326) and *MBOAT7* (rs641738) have been associated with the risk of NAFLD [15,16,17]. Not only these genes, but also other genetic variants related with obesity traits and loci have been associated with a higher risk of developing a severe stage of NAFLD [14,18]. 

Although these polymorphisms explain only a small fraction of the total heritability of NAFLD, it is possible that the combination of specific single nucleotide polymorphisms (SNPs) into a genetic risk score (GRS) could increase the detection and evolution of NAFLD [5,10,19]. Hence, some models have investigated the impact of the genetic predisposition to accumulate liver fat on NAFLD. In this context, some researchers have demonstrated an association between the combination of SNP in a GRS with de novo hepatocellular carcinoma (HCC) [16,20], but also with higher hepatic fat content, total cholesterol, steatosis degree and alanine aminotransferase (ALT) levels [5,15,17]. In this sense, an independent regulation of fat distribution from total adiposity has been suggested, where genes near loci regulating total body mass are enriched for expression in the Central Nervous System, while genes for fat distribution are enriched in adipose tissue itself [21]. 

Concerning treatment, there are no specific medications that directly treat NAFLD [22,23], being lifestyle modifications and weight control the most fundamental steps in the management of NAFLD [10]. In this context, the European Association for the Study of the Liver (EASL) recommends diet and physical activity as the best treatment for steatosis [24,25,26]. However, increasing evidence suggests that interindividual variability in weight loss also depends on interactions between genetic and environmental factors, including lifestyle [27,28,29]. Besides, the well and precise characterization of each gene and its different related pathways is essential in order to devise new therapies [6]. Indeed, a personalized treatment taking into account genetics, lifestyle and specific macronutrient recommendations is needed [26,30,31]. 

In this sense, the aim of the present study is to assess genetic and non-genetic factors putatively involved in the improvement of the hepatic health after a 6-month hypocaloric nutritional treatment. To test this hypothesis, we combined 47 obesity-related genetic variants associated with different NAFLD non-invasive methods based on the change of hepatic functionality (FLI), lipid metabolism (OWLiver^®^-test) and liver fat content (by MRI) into three differences scores, where the role of baseline status was assessed to predict outcomes from precise nutrition management.

## 2. Materials and Methods

### 2.1. Study Design

The current randomized controlled trial was designed to compare the effectiveness of two weight loss dietary strategies with different nutritional features, anthropometric measurements, body composition and biochemical markers on hepatic health in overweight or obese subjects with ultrasonography-proven liver steatosis, as described elsewhere [32,33]. The intervention had a duration of 24 months and the participants were randomly assigned to the American Heart Association (AHA) or the Fatty Liver in Obesity (FLiO) group [34]. However, the present study was performed concerning the results after 6 months of follow-up. The study was approved by the Research Ethics Committee of the University of Navarra, Spain on 24 April 2015 (ref. 54/2015) and accessed on www.clinicaltrials.gov (FLiO: Fatty Liver in Obesity study; NCT03183193). Each subject gave written informed consent prior to enrollment in the study. All the procedures were performed in accordance with the Declaration of Helsinki and the study was conducted following the CONSORT 2010 guidelines.

### 2.2. Study Participants

A total of 98 men and women with overweight or obesity (body mass index (BMI) ≥ 27.5 kg/m^2^ to < 40 kg/m^2^) between 40–80 years old and with hepatic steatosis confirmed by abdominal ultrasonography fulfilled the selection criteria and were enrolled in the study [35]. After 6 months, a total of 70 participants completed the evaluation. Exclusion criteria included the presence of known liver disease other than NAFLD, ≥3 kg of body weight loss in the last 3 months, excessive alcohol consumption (>21 standard drinks per week in men and >14 standard drinks per week for women) [36], endocrine disorders (hyperthyroidism or uncontrolled hypothyroidism), pharmacological treatments (immunosuppressants, cytotoxic agents, systemic corticosteroids or other drugs that could potentially cause hepatic steatosis or altering liver tests) [37], active autoimmune diseases or requiring pharmacological treatment, the use of weight modifiers and severe psychiatric disorders and the lack of autonomy or an inability to follow the diet, as well as the use of weight modifiers, severe psychiatric disorders and difficulties in following the scheduled visits.

### 2.3. Dietary and Lifestyle Intervention

Two energy-restricted diets, AHA (*n* = 41) and FLiO (*n* = 45), were prescribed [33]. Both diets applied an energy restriction of 30% of the total energy requirements of each participant in order to achieve a loss of at least 3–5% of the initial body weight, in accordance with the recommendations of the American Association for the Study of Liver Diseases guidelines (AASLD) [37]. After 6 months of nutritional treatment, both AHA and FLiO groups (*n* = 34 and 36, respectively) achieved comparable results in the evaluated main variables and no significant differences in the changes between the intervention groups were found [33]. Therefore, participants were merged and compared together. The habitual dietary intake was registered with a validated semi-quantitative food frequency questionnaire (FFQ) of 137 items, both at baseline and after the 6-month intervention [38,39]. The composition of the food items was derived from accepted Spanish food composition tables as previously described [4,40]. The adherence to the Mediterranean Diet was assessed with a validated 17-point score questionnaire [41,42]. A physical activity prescription of 10,000 steps/day was given to the participants [43,44]. The physical activity level was evaluated using the validated Spanish version of the Minnesota Leisure-Time Physical Activity Questionnaire [44]. The volume of activity was indicated in metabolic equivalent of the task (METs), as described elsewhere [43]. 

### 2.4. Anthropometric, Body Composition and Biochemical Assessments 

Anthropometric variables (body weight, height and waist circumference) and body composition (Lunar iDXA, Encore 14.5, Madison, WI, USA) were assessed in fasting conditions at the Metabolic Unit of the University of Navarra following standardized procedures [45]. BMI was calculated as the body weight divided by the squared height (kg/m^2^). Blood samples were properly collected after overnight fasting of 8–10 h and processed at the Laboratory of Biochemistry of the University of Navarra Clinic (CUN, Pamplona, Spain). Blood glucose, triglycerides (TG), aspartate aminotransferase (AST), alanine aminotransferase (ALT) and gamma-glutamyltransferase (GGT) concentrations were determined on a Cobas 8000 autoanalyzer with specific commercial kits and following the instructions of the company (Roche Diagnostics, Basel, Switzerland). Insulin, fibroblast growth factor 21 (FGF-21), leptin and adiponectin concentrations were quantified with specific ELISA kits (Demeditec; Kiel-Wellsee, Germany) in a Triturus autoanalyzer (Grifols, Barcelona, Spain). Insulin resistance was estimated using the Homeostasis Model Assessment Index (HOMA-IR), which was computed as HOMA-IR = (insulin (μU/mL) × glucose (mmol/L))/22.5 [4]. The Triglycerides/Glucose index (TyG) (ln[triglycerides (mg/dL) × glucose(mg/dL)/2)]) was also calculated as a surrogate of glucose tolerance [46].

### 2.5. Imaging Techniques for the Assessment of Liver Status

The whole liver evaluation was performed under fasting conditions at the University of Navarra Clinic. Liver steatosis was determined by ultrasonography (Siemens ACUSON S2000 and S3000, Erlangen, Germany) in accordance with previously described methodology [47]. The clinical classification was established according to a 4-point scale: less than 5% (grade 0), 5–33% (grade 1), 33–66% (grade 2) and greater than 66% (grade 3), as described elsewhere [48]. Finally, magnetic resonance imaging (Siemens Aera 1.5 T) was used to determine the hepatic volume and the fat content of the liver (Dixon technique) as reported by the manufacturer [32].

Fatty liver index (FLI) was calculated using serum triglycerides, BMI, waist circumference and GGT concentrations using the formula described elsewhere [12].

### 2.6. Metabolomics

The metabolomic test OWLiver^®^ (One Way Liver S.L., Bilbao, Spain) is a fasting blood probe able to measure the degree of NAFLD development [32]. The test score is based on a prospective study, where subjects had previously been diagnosed by liver biopsy [49]. The methodology of this test consisted of the measure of a panel of biomarkers that belong to the family of triacylglycerols (TGs), which are a reflection of the amount of fat and inflammation of the liver [32]. The final OWLiver^®^ score is generated by the relative metabolite concentrations, which are analyzed together in a specific algorithm that gives the probabilities of normal liver, steatosis or NASH. 

### 2.7. SNP Selection and Genotyping

A total of 86 oral epithelium samples were collected with a liquid-based kit (ORAcollect-DNA, OCR-100, DNA Genotek Inc, Ottawa, Canada). Genomic DNA was isolated using the Maxwell^®^ 16 Buccal Swab LEV DNA Purification Kit (Promega Corp, Madison, WI, USA). The quality characterization was carried out by dsDNA quantification (Qubit, Thermo Fisher, Waltham, MA, USA). A pre-designed panel of 95 genetic variants related to obesity and weight loss was applied and analyzed [10,50,51,52]. More information about these obesity-related SNPs can be found in a previous report [28]. Genotyping was performed by targeted next generation sequencing on Ion Torrent PGM equipment (Thermo Fisher Scientific Inc., Waltham, MA, USA) [53], as previously published [50,54]. Overall, the amplicon mean size was 185 bp. As quality control of the sequencing process, the number of readings per amplifier and per sample were doubly checked, to make sure there are more than 50× (in fact, it is above about 400× even the worst). Library construction was carried out using a custom-designed panel and the Ion 198 AmpliSeq Library Kit 2.0 (Thermo Fisher Scientific) as per the manufacturer’s protocol. The raw data were processed with the Ion Torrent Suite Server Version 5.0.4 (Thermo Fisher Scientific Inc, Waltham, MA, USA) using *Homo sapiens* (genome assembly Hg 19) as the reference genome for the alignment. A custom-designed Bed file was used to locate the SNPs of interest. Genetic variants were identified with the Torrent Variant Caller 5.0 (Thermo Fisher Scientific) with a minimum coverage value of 20. Hardy–Weinberg equilibrium, linkage disequilibrium and haplotype inferences were estimated using the Convert program (Version 1.31) and the Arlequin software (Version 3.0). Hardy–Weinberg equilibrium was calculated with a statistical test (Chi-square). 

### 2.8. Genetic Risk Score (GRS)

Three individual GRS based on the pre-designed panel of 95 SNPs were calculated for the change of each non-invasive diagnostic method (FLI, MRI and OWLiver^®^-test) (Figure 1) according to the following steps. Firstly, Kruskal–Wallis tests were performed to identify SNPs statistically or marginally associated with the change in FLI, liver fat content by MRI and metabolomics assessed by OWLiver^®^-test (absence of allele, presence of one allele or presence of two alleles) in our samples, obtaining a total of 47 SNPs with a *p*-value lower than 0.20. Secondly, post-hoc tests (Mann–Whitney U-test pairwise) were run to define differences between genotypes in order to be differentially coded as risk and non-risk groups with these 47 SNPs. A risk genotype was defined as the one that was associated with a lower change of FLI, liver fat content (MRI) and OWLiver^®^-test. Genotypes with similar effects were clustered in a single category. In a third step, Mann–Whitney U-test was applied to confirm statistical differences between the categorized genotype groups (risk vs. non-risk), selecting those SNPs showing at least a marginal statistical trend (*p* < 0.10) and excluding those with a low sample (<10%) in either category or due to collinearity. To evaluate the combined effects of the previously selected SNPs on the change of FLI, fat liver content and OWLiver^®^-test, the three individual GRS were calculated by summing the number of risk alleles at each locus [55,56].

### 2.9. Statistical Analysis

The primary outcome of the study was the weight loss, according to the current recommendations of the AASLD to ameliorate NAFLD features [37]. The sample size was estimated assuming a mean difference of weight loss of 1.0 (1.5 kg) between both dietary groups (AHA vs. FLiO) with a 95% confidence interval (α = 0.05) and a statistical power of 80% (β = 0.80). Considering a dropout rate of 20–30%, 50 subjects were included in each group of the study, even though two subjects were excluded from the AHA group due to the presence of important biochemical alterations in the initial assessment. This trial started with 98 participants but only 86 epithelium buccal cells from volunteers were available. Moreover, after 6 months, a total of 70 participants had complete information and epithelium buccal cells to carry out the study.

Results with normal distribution were expressed as means ± standard deviations (SD), whereas continuous skewed variables were presented as medians and interquartile ranges (IQR). Moreover, qualitative variables were expressed as number (n) and percentages (%). The normality of the distribution was checked through Shapiro–Wilk and Kolmogorov–Smirnov tests. Statistical differences for continuous variables at baseline (between men and women and according to age) and after the 6-month dietary intervention were estimated using Student’s *t*-tests of independent samples and Wilcoxon–Mann–Whitney (for non-normally distributed variables). Categorical variables were compared using a Chi-squared test.

Diagnostic tests of the regression assumption for linearity and equal variance of residuals, and the variance inflation factor (VIF) for testing collinearity between independent variables, were conducted. Multiple linear regression models were used to predict FLI, liver fat content (by MRI) and OWLiver^®^-test changes. All the designed GRS were used as continuous variables in the multiple linear regression models. In addition to genetic variants, other conventional factors of personalization were evaluated, including age, sex and the following variables at baseline: insulin (U/mL), FGF-21 (pg/mL) and protein (%), as well as potential interaction introducing the corresponding interaction terms to the models.

All *p*-values presented are two-tailed and were considered statistically significant at *p* < 0.05. Analyses were carried out using Stata version 12.1 software (StataCorp 2011, College Station, TX, USA).

## 3. Results

Baseline characteristics of the participants, including body composition, biochemical and nutritional characteristics, are reported separated by sex and age (Table 1). Overall, 57% (*n* = 49) of subjects were men. The average values of weight and waist circumference followed expected trends depending on sex. Triglycerides and insulin resistance-related variables (HOMA-IR and TyG) showed statistical differences between genders, being higher in men as compared to women. However, leptin concentration was significantly lower in males (20.1 ng/mL vs. 46.0 ng/mL in females). Analyzing variables associated with liver injury, statistical differences were observed in the fatty liver index (76.0 vs. 89.6) and in liver fat content measured by MRI (4.5% vs. 6.5%), showing worst hepatic health values in men in both measures. On the other hand, lipidomic analysis (OWLiver^®^-test) did not show significant differences. Concerning results according to age significant differences were only observed on weight, glucose and adiponectin concentrations and MedDiet Score. Regarding diet, the nutritional pattern of the study population was characterized by a relatively high consumption of energy derived from fat (37.4%), a concomitant low intake of carbohydrates (42.8%) and an average protein intake of 16.8%. Moreover, significant improvements in body composition, biochemical parameters, hepatic health variables, dietary intake and lifestyle factors were observed after the 6-month nutritional intervention, following the expected trends. 

To study the genetic risk association with NAFLD, of a total of 95 SNPs related to obesity, 47 genetic variants were chosen because they were statistically or marginally associated with the amelioration of the hepatic health measured by non-invasive NAFLD diagnostic methods (FLI, MRI and OWLiver^®^-test). Of those, 1 SNP was common among all methods: rs2959272 (*PPARG*). On the other hand, 30 SNPs were exclusively related to a specific method—17 for FLI rs1801133 (*MTHFR*), rs1055144 (*NFE2L3*), rs17817449 (*FTO*), rs8050136 (*FTO*), rs3751812 (*FTO*), rs9939609 (*FTO*), rs2075577 (*UCP3*), rs324420 (*FAAH*), rs1121980 (*FTO*), rs2419621 (*ACSL5*), rs1558902 (*FTO*), rs3123554 (*CNR2/FUCA1*), rs6567160 (*MC4R*), rs660339 (*UCP2*), rs2605100 (*LYPLAL1*), rs1800629 (*TNF*APROMOTOR), rs4994 (*ADRB3*); 3 for MRI: rs6861681 (*CPEB4*), rs1440581 (*PPM1K*), rs1799883 (*FABP2*); and 10 for OWLiver^®^ test: rs1175544 (*PPARG*), rs1797912 (*PPARG*), rs1386835 (*PPARG*), rs709158 (*PPARG*), rs1175540 (*PPARG*), rs1801260 (*CLOCK*), rs12502572 (*UCP1*), rs8179183 (*LEPR*), rs894160 (*PLIN1*), rs4731426 (*LEP*). In our population, we used the SNP associated with each non-invasive method for calculating the GRS (Appendix A). 

The GRS, calculated as the number of risk alleles carried by each subject, was normally distributed. The sample was stratified, by the median, into a “low genetic risk group” (those with a GRS_FLI_ ≤ 9, GRS_OWL_ ≤ 10 and GRS_MRI_ ≤ 4 risk alleles) and a “high genetic risk group” (those with a GRS_FLI_ > 9, GRS_OWL_ > 10 and GRS_MRI_ > 4 risk alleles). The results for the effect of each GRS on the change of different outcomes after the nutritional treatment are shown in Table 2. All groups exhibited a significant body weight loss, which was higher when the genetic risk was lower. Moreover, body composition variables including weight, BMI and waist circumference showed statistical differences when comparing GRS_FLI_ and GRS_MRI_ medians. Furthermore, general improvements in biochemical parameters were found. However, the amelioration was only statistically significant in TG, TyG and leptin concentrations, and TG and FGF-21 concentrations, when comparing GRS_FLI_ and GRS_MRI_ medians, respectively. On the other hand, no significant changes were found for dietary intake and lifestyle factors. 

In order to evaluate the improvement of hepatic health depending on genetic and non-genetic risk factors, linear regression models were constructed (Table 3). These models were adjusted for sex, age, baseline protein intake, baseline FGF-21 and insulin concentration and the change in MedDiet Score. All the GRS included in the models showed an important association with the improvement on hepatic health. Moreover, a higher decrease in FLI was significantly associated with baseline insulin and protein and with the change in MedDietScore, showing an improvement in hepatic functionality. A high intake of protein at baseline also seemed to be important in the improvement of lipid metabolism, assessed by OWLiver^®^-test. Parallelly, the regression model established for the change in liver fat content (MRI) showed a significant interaction between the GRS_MRI_ and protein intake at baseline (*p*-value: 0.001). However, no statistically significant interactions between GRS_FLI_ and GRS_OWL_ were found. Overall, the change of FLI, OWLiver^®^-test and MRI variabilities were explained in approximately 52% (adj. R^2^ = 0.53), 16% (adj. R^2^ = 0.16) and 34% (adj. R^2^ = 0.34), respectively. 

In addition, Figure 2A–C plot simple linear regression analyses of statistically significant predictors of FLI, OWLiver^®^-test and liver fat content (MRI) decrease by diet. A lower change in FLI and OWLiver^®^-test was associated with a higher baseline protein intake (*p*-value: 0.009 and 0.022, respectively). Moreover, this association became more important when the genetic risk was higher (Figure 2A,B). Figure 2C shows that a higher baseline protein was associated with a lower change of liver fat content (by MRI) becoming the effect more evident when the genetic risk was higher (*p* interaction: 0.017).

## 4. Discussion

NALFD has reached pandemic levels, being recognized as an important health burden with an urgent need for early diagnosis [57]. Genetic predisposition for NAFLD has been reported [18,20]. With this in mind, the objective of this research was to assess the impact of the interaction between genetic and non-genetic factors concerning the improvement of the hepatic health using different diagnosis tools (FLI, MRI and OWLiver^®^-test) after a 6-month energy-restricted nutritional treatment for a more personalized management of this liver disease.

The high prevalence of NAFLD could be related to its strong link with obesity, which seems to play a role in both the initial simple steatosis and in its progression to NASH [11,58]. In this context, various genes and less frequent variants have been associated with the regulation of energy metabolism [18,59]. Moreover, the increasing of knowledge of the genetic component of NAFLD has promoted the development of noninvasive diagnosis methods based on Genome Wide Associations Studies (GWAS) [11,16,57,60], but few of them have examined the contribution of obesity-related variants linked to the evolution of this hepatic disease [37,61].

In order to better understand the contribution of genetics in the context of NAFLD, three different GRS were constructed based on the improvement of hepatic health after an energy-restricted treatment measured by three non-invasive diagnostic methods (FLI, magnetic resonance imaging and OWLiver^®^-test). On the one hand, fatty liver index has been highly correlated with measures of fatty liver disease showing an area under the curve of 0.84, predicting most cases of NAFLD [62]. Moreover, a recent study reported that the FLI joint to the waist circumference-to-height ratio could be one of the most accurate algorithms for the noninvasive diagnosis of NAFLD in both lean and overweight/obese population [63]. On the other hand, MRI can be considered the gold standard for steatosis measurement, being highly accurate and reproducible and superior in detecting and quantifying fat accumulation [61,64]. However, these two methods have limitations in detecting inflammation, ballooning and cellular injury, which are key components in NASH diagnosis [65]. Thus, in some cases, models based on “omics” sciences, such as the OWLiver^®^-test, could be of interest adding knowledge about diverse factors influencing weight loss variability among individuals. Due to the differences between methods in outcome measures, distinct genes and so pathways may be expected to be connected.

Therefore, a total of 47 polymorphisms were independently associated with differential responses to hepatic functionality (FLI), fat liver content (MRI) and lipid metabolism (OWLiver^®^-test). It is important to emphasize that each non-invasive diagnostic method has its specific SNPs. Only the rs2959272 (*PPARG*) genetic variant was the common element on the three GRS. In this sense, an intervention study indicated that the PPARG genotype was associated with success in body weight reduction [66]. Indeed, two common elements were also observed between GRS_FLI_ and GRS_MRI_, SNPs were located in genes related to bile secretion (*ABCB11*) and the regulation of energy balance and body weight (*SH2B1*). Meanwhile, SNPs in genes implicated in weight loss (*SH2B1* and *STK33*) influenced both GRS_OWL_ and GRS_MRI_. Instead, three common elements mapped to genes involved in endocrine/enzymatic regulation of lipid metabolism affecting macronutrient (*GNAS*) food intake and energy expenditure (*MC4R*) and thermogenesis (*UCP1*) were observed in GRS_FLI_ and GRS_OWL_.

In this study, a greater change in most of the NAFLD-related variables was reported when the genetic risk was lower. According to these findings, it has been extensively debated the identification of the physiological pathways that control energy metabolism and body weight regulation [31,67]. A Genetic Investigation of ANtropometric Traits consortium (GIANT) metanalysis identified 97 loci for BMI where genes near these specific loci showed expression enrichment in the central nervous system, suggesting that BMI is mainly regulated by processes such as hypothalamic control of energy intake [68]. Similar results have been found in a recent study in a pediatric population, where the application of a GRS to established clinical risk factors significantly improved the discriminatory capability of the prediction of NAFLD risk [5]. Indeed, different genetic variants and interactions with environmental factors have been shown to modulate the differential individual responses to moderately high-protein and low-fat dietary interventions in a Caucasian population [55]. In this sense, genetic information could help to determine the most appropriate dietary intervention for the prevention and treatment of NAFLD, as well as the development of associated comorbidities [69].

Moreover, for the purpose of explaining the variability on the improvement in hepatic functionality (FLI), liver fat content (by MRI) and lipidomic (OWLiver^®^-test), linear regression models were performed. The predictive accuracy of all models substantially improved when combining each of the previously mentioned SNPs in the multiple linear regression models, which is in line with previous studies [70]. In order to ameliorate these results, each regression model was fitted by sex, age and NAFLD-related variables such as inflammatory biomarkers or dietary compounds. Other variables such as the nutritional group of the participants was also considered, even though no significant differences were found. Factors related with proinflammatory and profibrogenetic pathways such as leptin, adiponectin or FGF-21, which appears to be elevated in patients with NAFLD, are therefore a promising target for the treatment [71,72]. Thus, GRS_FLI_, GRS_MRI_ and GRS_OWL_ were major predictors of the change in FLI, liver fat content (MRI) and OWLiver^®^-test, respectively.

To the best of our knowledge, there are few studies showing the combined effects of GRS built from SNPs related weight and adiposity regulation in response to different energy-restricted diets [50]. Moreover, it has been reported that genetic background is an important factor explaining metabolically health and unhealthy phenotypes related to obesity, in addition to lifestyle variables [54].

In this sense, dietary factors seem to be of key importance and have been associated with weight gain, obesity and NAFLD development [73,74]. Interestingly, in this research, higher baseline protein was associated with worst hepatic health improvement measured by FLI and OWLiver^®^-test. Furthermore, an interaction between the liver fat content assessed by MRI and baseline protein was found. In the same line were the results obtained from the Nurse´s Health Study and the Health Professionals Follow-up Study, where an increased intake of sugar-sweetened beverages was found to amplify the association of a 32-SNP genetic risk score with BMI [75]. These findings suggest that not only genetic and dietary factor should be considered but also the interaction of both of them [76,77]. Hence, a combined analysis over 16,000 children and adolescent showed the *FTO* rs9939609 variant that confers a predisposition to higher BMI is associated with higher total energy intake and that lower dietary protein intake attenuates the association [78]. Among the macronutrient categories, protein is the main one that contributes to the satiety, therefore contributing to weight loss [79,80]. However, the effect of the high protein diet in patients with NAFLD remains controversial. On the one hand, it has been suggested that the consumption of specific dietary amino acids might negatively impact liver status and, to a lesser extent, glucose metabolism in subjects with overweight/obesity and NAFLD [81]. Moreover, high protein intake derived mainly from dairy products has been associated with higher risk to develop diabetes and also with NAFLD [82]. A recent study has also suggested that following a lower protein diet, particularly in genetically predisposed individuals, might be an effective approach for addressing cardiometabolic diseases among Southeast Asian women [83]. On the other hand, high protein diet has been reported to be a valid therapeutic approach to revert NAFLD, being of special importance the protein source and the functional status of the liver [84]. In addition, BCAA supplementation has been demonstrated to ameliorate liver fibrosis and suppress tumor growth in a rat model of HCC with liver cirrhosis, as well as alleviate hepatic steatosis and liver injury in NASH mice [85,86].

There are some drawbacks of this research. Firstly, liver biopsy results were not available to corroborate the precise diagnosis of patients [57]. Nonetheless, in this research, we carried out a complete evaluation of the liver status by means of validated and widely used techniques as well as blood biomarkers and hepatic indexes, which are affordable and practical methods to use in health assessment. Second, the sample size and the enrollment of subjects are not very large. For this reason, these models may be further validated in different populations to establish whether they might represent a reliable and accurate, “non-invasive alternative” to liver biopsy. In addition, the role of new SNPs associated with excessive adiposity and accompanying metabolic alterations through a GRS approach needs to be re-explored, but our contribution re-states the value of the genetic make-up when prescribing personalized diets. Thirdly, type I and type II errors cannot be completely ruled out, especially those related to the selection of SNPs to be introduced into the GRS. However, due to the use of less stringent *p*-value thresholds compared to association studies of single variants, genomic profile risk scoring analyses can tolerate, at balance, some of these biases, as previously reviewed [87]. Fourthly, dietary intake was evaluated using self-reported information of the participants, which may produce some bias on the evaluation of the results. Lastly the constructions of the GRS using specific obesity-related SNP it is also an important limitation. However, the inclusion of these SNP on the evaluation of the genetic influence on NAFLD could be also considered an important strength of this investigation, as well as the use of different multiple linear regression models to test the contribution of genetics, baseline protein and inflammatory factors on the management of NAFLD. Finally, the study is a randomized controlled trial where each volunteer had an individual follow-up promoting the adoption of behavioral changes and a healthy lifestyle.

Overall, this experiment was designed as a proof of concept in order to evaluate if the genetic background linked to NAFLD-related factors may influence hepatic amelioration. In addition, examining new causes of disease and the underlying mechanism or alteration in specific pathways and clinical outcomes may be of interest.

## 5. Conclusions

Predicting the individual risk of NAFLD and determining the probability of disease progression is the basis for a precise diagnosis and treatment. These results demonstrate that three different genetic scores can be useful for the personalized management of NAFLD, whose treatment must rely on specific dietary recommendations guided by the measurement of specific genetic biomarkers.

## Figures and Tables

**Figure 1 diagnostics-11-01083-f001:**
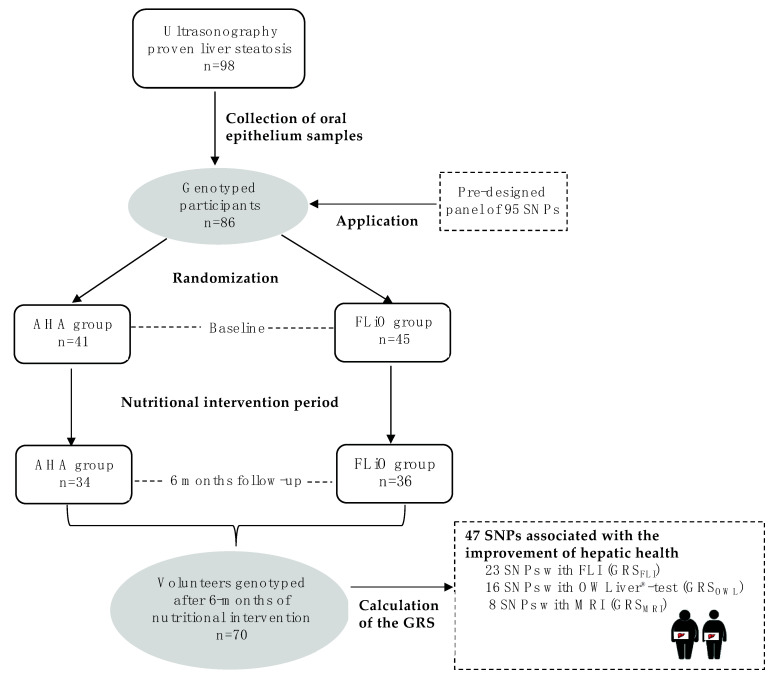
Design and flow chart of participants in the Fatty Liver in Obesity (FLiO) study. AHA, American Heart Association; FLI, fatty liver index; FLiO, Fatty Liver in Obesity; GRS, genetic risk score; GRSFLI, genetic risk score for FLI; GRSMRI, genetic risk score for magnetic resonance imaging; GRSOWL, genetic risk score for OWLiver^®^-test; MRI, magnetic resonance imaging; SNPs, Single Nucleotide Polymorphisms.

**Figure 2 diagnostics-11-01083-f002:**
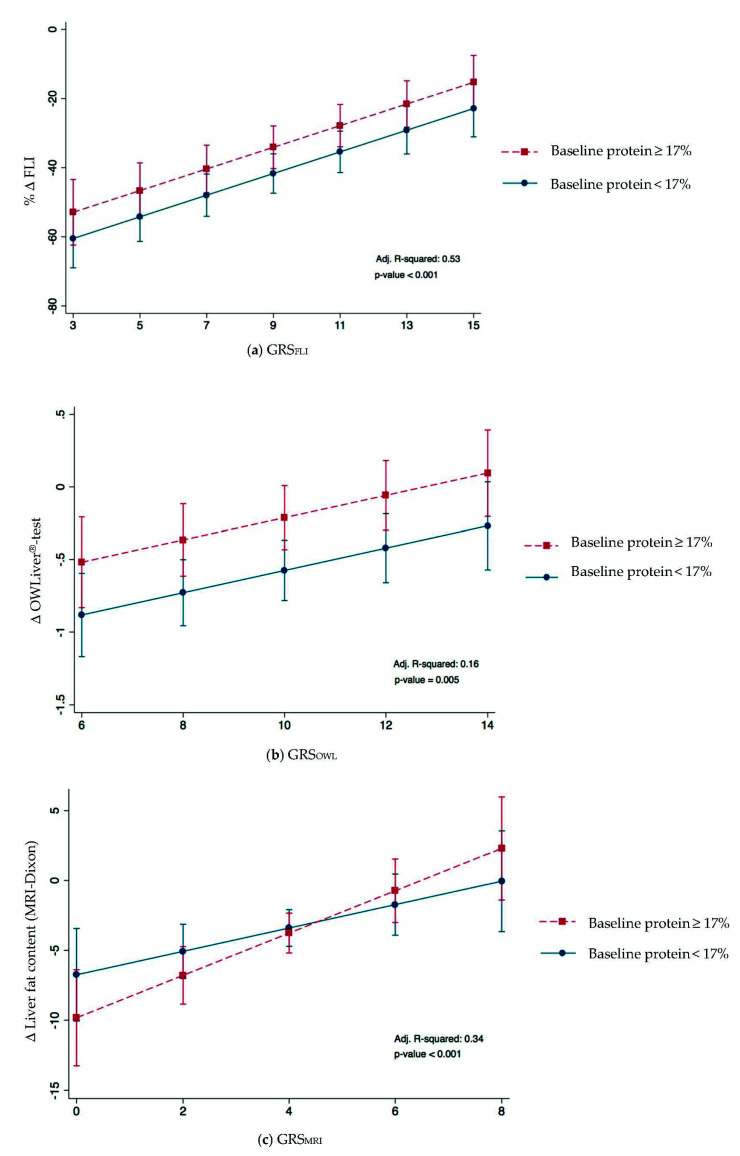
Effect of the changes in (**a**) % FLI and GRS_FLI_, (**b**) OWLiver^®^-test and GRS_OWL_ and (**c**) MRI and GRS_MRI_ and baseline protein after 6-month nutritional treatment. Baseline protein was dichotomized according to median.

**Table 1 diagnostics-11-01083-t001:** Baseline characteristics and after 6 months of dietary intervention and according to sex and age.

	Baseline					
	All Participants	Women	Men	≤50 y	>50 y	6 Months ª
n	86	37	49	43	43	70
Body composition						
Weight (kg)	95.0 (13.9)	88.1 (13.0)	100.3 (12.3) ***	98.5 (13.8)	91.6 (13.3) *	84.4 (75.1; 92.1) ***
BMI (kg/m^2^)	32.8 (30.6; 35.8)	32.9 (30.0; 36.2)	32.5 (30.9; 35.8)	33.8 (30.9; 36.6)	32.2 (30.2; 34.6)	28.7 (27.6; 32.6) ***
WC (cm)	109.0 (9.1)	103.8 (7.4)	112.9 (8.3) ***	109.2 (8.7)	108.7 (9.6)	99.9 (9.5) ***
DXA VAT (kg)	2.3 (1.6; 3.1)	2.2 (1.5; 3.1)	2.3 (1.7; 3.1)	2.3 (1.7; 3.2)	2.3 (1.6; 3.0)	1.5 (0.7) ***
Biochemical parameters						
TG (mg/dL)	121.0 (80.0; 155.0)	92.0 (72.0; 133.0)	127.0 (96.0; 170.0) **	126.0 (80.0; 167.0)	113.0 (72.0; 148.0)	83.0 (56.0; 114.0) ***
Glucose (mg/dL)	101.5 (91; 108)	97.0 (92.0; 105.0)	103.0 (91.0; 115.0)	97.0 (91.0; 104.0)	104.0 (95.0; 112.0) *	92.0 (87.0; 98.0) ***
Insulin (U/mL)	16.6 (7.9)	15.3 (7.8)	17.7 (7.9)	15.4 (7.1)	17.9 (8.5)	8.6 (6.5; 13.5) ***
HOMA-IR	4.1 (2.8; 5.7)	3.5 (2.2; 5.1)	4.4 (3.1; 6.0) *	3.9 (2.6; 5.1)	4.4 (2.9; 6.5)	1.9 (1.4; 3.3) ***
TyG index	8.6 (0.5)	8.4 (0.4)	8.8 (0.4) ***	8.6 (0.4)	8.6 (0.5)	8.2 (7.8; 8.6) ***
Adiponectin (μg/mL)	6.3 (5.0; 8.3)	6.6 (5.2; 8.3)	5.9 (5.0; 7.5)	5.8 (4.5; 7.9)	6.6 (5.7; 9.7)	8.0 (6.1; 9.9) ***
Leptin (ng/mL)	29.8 (17.6; 44.8)	46.0 (37.7; 69.3)	20.1 (14.0; 26.4) ***	32.1 (17.6; 46.09)	26.4 (15.9; 39.3) *	16.7 (7.5; 33.0) ***
FGF21 (pg/mL)	211.5 (108.0; 352.0)	190.0 (89.1; 387.0)	215.0 (124.0; 328.0)	182.0 (87.7; 302.0)	244.0 (130.0; 416.0)	187.5 (111.0; 355.0)
Liver injury						
FLI	83.1 (73.7; 92.3)	76.0 (60.7; 83.0)	89.6 (79.8; 94.1) ***	84.4 (74.2; 93.3)	79.8 (70.5; 91.7)	51.11 (23.6) ***
MRI Liver fat—Dixon (%)	5.6 (3.2; 9.6)	4.5 (2.9; 8.7)	6.5 (4.3; 10.1) *	5.9 (3.5; 12.4)	5.0 (3.0; 8.9)	2.0 (1.3; 3.8) ***
Lipidomic (OWLiver^®^-test) n (%)						
No NAFLD	17 (20.0)	7 (18.9)	10 (20.8)	7 (16.2)	10 (23.8)	23 (32.8)
Hepatic Steatosis	20 (23.5)	10 (27.0)	10 (20.8)	7 (16.2)	13 (30.9)	21 (30.0) *
NASH	48 (56.4)	20 (54.0)	28 (58.3)	29 (67.4)	19 (45.2)	26 (37.1)
Dietary intake per day						
Total energy (kcal/day)	2550 (1958; 2925)	2548 (2031; 3133)	2554 (1897; 2902)	2551 (2042; 3066)	2464 (1833; 2864)	2004 (576) ***
Carbohydrates (%E)	42.8 (37.6; 47.8)	43.0 (35.9; 48.4)	42.5 (39.2; 47.5)	40.8 (36.2; 46.8)	43.0 (39.9; 48.0)	42.3 (7.7)
Proteins (%E)	16.8 (15.1; 19.1)	16.9 (15.2; 20.9)	16.7 (15.1; 19.0)	16.7 (15.3; 18.8)	16.9 (14.6; 19.3)	19.4 (17.1; 22.8) ***
Fats (%E)	37.4 (6.8)	38.1 (7.5)	36.8 (6.2)	38.3 (6.8)	36.5 (6.7)	35.4 (7.8)
Lifestyle factors						
MedDiet Score	5.9 (1.9)	5.9 (2.3)	5.9 (1.6)	5.4 (1.7)	6.4 (2.0) *	12.0 (10.0; 14.0) ***
PA (METs-min/week)	2240 (1665; 4307)	2240 (1710; 4307)	2280 (1100; 4365)	2322 (1705; 4365)	2216 (1392; 4200)	3720 (2442; 5115) ***

Variables re shown as mean (SD) or as median (IQR) according to its distribution. Categorical variables are presented as absolute (*n*). Paired *t*-tests and Wilcoxon-matched-pairs signed ranks were carried out to compare baseline and 6 months participants characteristics. Independent samples *t*-tests and Wilcoxon–Mann–Whitney were carried out to compare changes between sex and age groups. Age was categorized according to the median. * *p* < 0.05; ** *p* < 0.01; *** *p* < 0.001. ª Comparison within dietary groups (baseline and after 6 months). BMI, body mass index; DXA, dual-energy X-ray absorptiometry; %E, percentage of energy; FGF-21, fibroblast growth factor 21; FLI, fatty liver index; HOMA-IR, homeostasis model assessment insulin resistance; MRI, magnetic resonance imaging; OWL^®^; OWLive^r®^-test; PA, physical activity; TG, triglycerides; TyG index, triglycerides and glucose index; VAT, visceral adipose tissue; WC, waist circumference.

**Table 2 diagnostics-11-01083-t002:** Change in body composition, biochemical, dietary and lifestyle factors according to different genetic risk scores.

	GRS_FLI_		GRS_OWL_		GRS_MRI_	
	<9	≥9	<10	≥10	<4	≥4
n	30	40	29	41	31	39
Body composition						
Mean	6.0 (1.6)	9.6 (3.9)	7.0 (1.5)	11.7 (1.4)	2.0 (0.8)	4.9 (1.0)
ΔWeight (kg)	−10.6 (−15.8; −6.9)	−8.4 (−10.7; −5.0) *	−9.5 (−12.4; −6.4)	−8.6 (−12; −6.4)	−9.6 (−16.9; −8.2)	−7.6 (−11.2; −4.9) *
ΔBMI (kg/m^2^)	−3.6 (−5.1; −2.4)	−2.9 (−3.7; −1.6) *	−3.3 (−4.7; −2.2)	−3.2 (−4.1; −2.2)	−3.5 (−5.6; −2.9)	−2.8 (−3.8; −1.6) *
ΔWC (cm)	−11.7 (6.5)	−7.6 (5.6) **	−9.1 (6.4)	−9.5 (6.3)	−11.0 (5.2)	−8.0 (6.8) *
ΔDXA VAT (kg)	−0.8 (−1.2; −0.4)	−0.9 (−1.5; −0.4)	−1.0 (−1.5; −0.4)	−0.8 (−1.5; −3.2)	−1.0 (−1.5; −0.6)	−0.7 (−1.67; −0.2)
Biochemical parameters						
ΔTG (mg/dL)	−42.0 (−100.0; −18.0)	−15.0 (−56.0; 4.0) *	−32.0 (−68; 0)	−22.5 (−58.5; −0.5)	−46.0 (−102.0; −5.0)	−18.0 (−45.0; 1.0) *
ΔGlucose (mg/dL)	−8.6 (11.0)	−10.2 (12.4)	−9.8 (13.6)	−9.3 (10.4)	−9.2 (11.0)	−9.8 (12.5)
ΔInsulin (U/mL)	−7.4 (7.1)	−4.9 (8.0)	−6.2 (8.7)	−5.9 (7.0)	−6.7 (7.4)	−5.4 (8.0)
ΔHOMA-IR	−2.0 (−3.2; −0.2)	−1.6 (−3.3; −0.2)	−1.9 (−3.2; −0.2)	−1.8 (−3.2; −0.6)	−2.3 (−3.2; −1.0)	−1.5 (−3.2; −0.1)
ΔTyG index	−0.6 (0.4)	−0.2 (0.4) **	−0.4 (0.3)	−0.4 (0.5)	−0.5 (0.5)	−0.3 (0.3)
ΔAdiponectin (μg/mL)	1.5 (0.1; 4.7)	1.2 (−0.9; 3.2)	−0.1 (−0.6; 2.2)	1.8 (0.1; 3.4)	1.0 (−0.6; 3.4)	1.3 (−0.2; 4.1)
ΔLeptin (ng/mL)	−11.1 (−21.6; −7.2)	−7.5 (−14.5; −2.9) *	−9.5 (−15.8; −7.0)	−7.5 (−20.0; −3.0)	−9.1 (−15.8; −6.7)	−9.1 (−20.0; −3.0)
ΔFGF21 (pg/mL)	−9.1 (−123.0; 80.0)	−40.5 (−146.5; 95.5)	−41.7 (−132; 50)	−0.8 (−123; 88)	−55.4 (−217; 45)	12.0 (−64.2; 97) *
ΔFLI (%)	−54.5 (19.7)	−22.6 (17.9) ***	−33.7 (21.8)	−38.4 (26.2)	−41.1 (25.3)	−32.9 (23.5)
ΔMRI Liver fat—Dixon (%)	−2.7 (−6.8; −0.7)	−2.7 (−6.8; −1.2)	−4.3 (−8; −0.8)	−3.4 (−6.8; −1.2)	−4.5 (−7.8; −2.5)	−1.6 (−4.2; −0.2) ***
**Δ** **Lipidomic (OWLiver^®^-test) n (%)**						
OWL^®^ maintenance	19 (63.3)	28 (71.7)	13 (44.8)	34 (85.0) ***	17 (56.6)	30 (76.9)
OWL^®^ reduction	11 (36.6)	11 (28.2)	16 (55.1)	6 (15.0)	13 (43.3)	9 (23.0)
Dietary intake per day						
ΔTotal energy (kcal)	−882 (−1261; −88)	−523 (−1099; −101)	−589 (−987; −132)	−603 (−1175; 44)	−881 (−1257; −308)	−479 (−1009; 66)
ΔCarbohydrates (%)	−1.3 (10.0)	−0.7 (8.7)	−1.9 (8.2)	−0.3 (9.9)	−2.0 (9.4)	−0.3 (9.1)
ΔProteins (%)	3.6 (4.3)	1.9 (5.7)	2.5 (3.5)	2.7 (6.1)	2.9 (6.0)	2.4 (4.6)
ΔFats (%)	−0.7 (−6.2; 4.5)	−2.1 (−9.1; 5.1)	−0.2 (−5.2; 5.2)	−1.8 (−10.1; 4.8)	−2.6 (−8.3; 4.1)	−1.4 (−9.8; 5.8)
Lifestyle factors						
ΔMedDiet Score	6.3 (3.1)	5.7 (3.4)	5.7 (3.2)	6.1 (3.4)	6.5 (3.3)	5.5 (3.3)
ΔPA (METs min/week)	758 (−217; 2405)	1215 (−120; 2798)	896 (73; 2798)	1111 (−753; 2405)	984 (−65; 2357)	1046 (−557; 2817)

Variables are shown as mean (SD) or as median (IQR) according to its distribution. Categorical variables are presented as absolute (*n*). Independent samples *t*-tests and Wilcoxon–Mann–Whitney were carried out to compare variables changes according to the median of GRS_FLI_ < 9 and GRS_FLI_ ≥ 9, GRS_MRI_ < 4 and GRS_MRI_ ≥ 4 and GRS_OWL_ < 10 and GRS_OWL_ ≥ 10. * *p* < 0.05; ** *p* < 0.01; *** *p* < 0.001. BMI, body mass index; DXA, dual-energy X-ray absorptiometry; %E, percentage of energy; FGF-21, fibroblast growth factor 21; FLI, fatty liver index; HOMA-IR, homeostasis model assessment insulin resistance; MRI, magnetic resonance imaging; OWL^®^; OWLiver^®^-test; PA, physical activity; TG, triglycerides; TyG index, triglycerides and glucose index; VAT, visceral adipose tissue; WC, waist circumference.

**Table 3 diagnostics-11-01083-t003:** Linear regression analyses of changes in fatty liver index, OWLiver^®^-test and liver fat content (MRI).

		β	*p*-Value	Adjusted R^2^	*p*-Model
% Change in Fatty Liver Index (FLI)					
Model 1	GRS_FLI_	3.75	<0.001	0.37	<0.001

Model 2	GRS_FLI_	3.37	<0.001	0.39	<0.001
	Baseline protein	1.27	0.044		

Model 3	GRS_FLI_	3.03	<0.001	0.45	<0.001
	Baseline protein	1.55	0.011		
	Baseline insulin	0.80	0.005		

Model 4	GRS_FLI_	3.10	<0.001	0.53	<0.001
	Baseline protein	1.49	0.009		
	Baseline insulin	0.76	0.005		
	Change MedDiet Score	−1.99	0.002		
Change in OWLiver^®^-test					
Model 5	GRS_OWL_	0.08	0.001	0.12	0.009

Model 6	GRS_OWL_	0.07	0.011	0.16	0.005
	Baseline protein	0.04	0.022		
Change in liver fat content (MRI)					
Model 7	GRS_MRI_	1.13	<0.001	0.23	<0.001

Model 8	GRS_MRI_	1.28	<0.001	0.24	<0.001
	Baseline protein	0.04	0.741		

Model 9	GRS_MRI_	1.17	<0.001	0.28	<0.001
	Baseline protein	0.09	0.448		
	Baseline FGF21	−0.004	0.051		

Model 10	GRS_MRI_#baselineprotein	0.180	0.017	0.34	<0.001
	Baseline FGF21	−0.004	0.040		

All models were adjusted by age and sex. GRS_FLI_, genetic risk score for FLI; GRS_MRI_, genetic risk score for magnetic resonance imaging; GRS_OWL_, genetic risk score for OWLiver^®^-test; MRI, magnetic resonance imaging.

## Data Availability

The data presented in this study are available on request from the corresponding author.

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
