# Peer review of "Three Different Genetic Risk Scores Based on Fatty Liver Index, Magnetic Resonance Imaging and Lipidomic for a Nutrigenetic Personalized Management of NAFLD: The Fatty Liver in Obesity Study"

_diagnostics, 2021, doi:10.3390/diagnostics11061083_

Round 1

Reviewer 1 Report

Manuscript ID: diagnostics-1222244

Title: "Three different genetic risk scores based on Fatty Liver Index, Magnetic Resonance Imaging and lipidomic for a nutrigenetic personalized management of NAFLD: the Fatty Liver in Obesity study".

Authors: Nuria Perez-Diaz-del-Campo, et al.

The authors of the present manuscript tried to identify and evaluate the potential genetic or non-genetic factors associated with the development and progression of NAFLD in overweight or obese patients following two different weight-loss dietary treatments. It is an overall interesting study focusing on an emerging medical field and a significant health disorder.

Major comments:

  1. According to the authors: "NAFLD is defined as fatty liver without injury of the hepatocyte in the form of ballooning, with or without inflammation". Could the authors clarify whether this definition corresponds to NAFL? Do they also use the term NAFLD to describe the entire spectrum of the disease?
  2. Why did the authors decide to perform the analysis after six months of intervention? Do they have any available information after the study was completed?
  3. How did the authors exclude the presence of NASH using abdominal ultrasonography at the study enrollment?
  4. Would it be possible for the authors to clarify the sentence "After 6 months a total of 76 participants completed the evaluation."? Do they mean the dietary intervention that is described later?
  5. It would be helpful if the authors could mention the specific time points they performed their assessments in the material and methods section.
  6. It would be useful if the authors could provide a brief description of the quality control steps they performed in their genetic analysis. Moreover, did they control for population stratification?
  7. The authors mention that "The normality of the distribution was checked through Shapiro-Wilk and Kolmogorov–Smirnov test." Can they clarify this point? Did they require both tests to confirm normal distribution?
  8. Did the authors consider correcting for multiple tests?
  9. Did the authors adjust their results based on which nutritional group the participants belong to?
  10. It would be helpful if the authors could mention in table 1 how many of the participants were included in each nutritional group.

Author Response

We would like to thank the reviewer for their thoughtful comments and suggestions for improvement.
We believe that in addressing these comments, this revised manuscript is considerably ameliorated.
Changes in the manuscript have been marked up using the “Track Changes” and are shown in this
document as well. A detailed point-by-point response to the reviewer is provided after each comment.
RESPONSE TO REVIEWER #1 COMMENTS
The authors of the present manuscript tried to identify and evaluate the potential genetic or non-genetic
factors associated with the development and progression of NAFLD in overweight or obese patients
following two different weight-loss dietary treatments. It is an overall interesting study focusing on an
emerging medical field and a significant health disorder.
Major comments:
1. According to the authors: "NAFLD is defined as fatty liver without injury of the hepatocyte in
the form of ballooning, with or without inflammation". Could the authors clarify whether this
definition corresponds to NAFL? Do they also use the term NAFLD to describe the entire
spectrum of the disease?
Thank you for your comments. We have rewritten this statement after considering the reviewer
recommendation and specifying as follows:
1. Introduction (Page 2 line 69-72)
NAFLD includes a spectrum of liver diseases conditions ranging from simple steatosis to non-alcoholic
steatohepatitis (NASH) with variable degrees of fibrosis and cirrhosis 2,3, and it has become one of the
most common causes of chronic liver diseases.
2. Koch, L. K.; Yeh, M. M. Nonalcoholic Fatty Liver Disease (NAFLD): Diagnosis, Pitfalls, and Staging. Ann. Diagn. Pathol. 2018, 37,
83–90. https://doi.org/10.1016/j.anndiagpath.2018.09.009.
3. Pais, R.; Maurel, T. Natural History of NAFLD. J. Clin. Med. 2021, 10 (6), 1161. https://doi.org/10.3390/jcm10061161.
2. Why did the authors decide to perform the analysis after six months of intervention? Do
they have any available information after the study was completed?
Thank you for this interesting question. We decided to performed the analysis after 6-months of
nutritional intervention because this period is enough to observed significant different effects and
analyzing possible nutrigenetic interactions, as it has been shown in other studies with similar designs [1-
3].
1. Surendran, S., Aji, A. S., Ariyasra, U., Sari, S. R., Malik, S. G., Tasrif, N., Yani, F. F., Lovegrove, J. A., Sudji, I. R., Lipoeto, N. I., &
Vimaleswaran, K. S. (2019). A nutrigenetic approach for investigating the relationship between vitamin B12 status and
metabolic traits in Indonesian women. Journal of diabetes and metabolic disorders, 18(2), 389–399.
https://doi.org/10.1007/s40200-019-00424-z
2. Franck M, de Toro-Martín J, Guénard F, Rudkowska I, Lemieux S, Lamarche B, Couture P, Vohl MC. Prevention of Potential
Adverse Metabolic Effects of a Supplementation with Omega-3 Fatty Acids Using a Genetic Score Approach. Lifestyle Genom.
2020;13(1):32-42. https://doi.org/10.1159/000504022
3. Goni L, Cuervo M, Milagro FI, Martínez JA. A genetic risk tool for obesity predisposition assessment and personalized nutrition
implementation based on macronutrient intake. Genes Nutr. 2015 Jan;10(1):445. https://doi.org/10.1007/s12263-014-0445-
z.
Indeed, in this specific cohort, after six months of intervention participants we achieved relevant clinical
outcomes concerning weight-loss and hepatic metabolism [Reference 33 in the manuscript], which
support the validity of our trial.
33. Marin-Alejandre, B. A.; Abete, I.; Cantero, I.; Monreal, J. I.; Martinez-echeverria, A.; Uriz-otano, J. I. The Metabolic and Hepatic
Impact of Two Personalized Dietary Strategies in Subjects with Obesity and Nonalcoholic Fatty Liver Disease : The Fatty Liver
in Obesity ( FLiO ) Randomized Controlled Trial. Nutrients 2019, 11 (10). https://doi.org/10.3390/nu11102543.
3. How did the authors exclude the presence of NASH using abdominal ultrasonography at the
study enrollment?
We thank for this comment. As reviewer indicates ultrasonography has some limitations such as being
too simple to detect small changes in steatosis severity on follow up [1-2]. In this study, ultrasonography
was used only for scaling the selection of volunteers, while the quantification of liver fat was assessed by
Magnetic Resonance Imaging, as described in the manuscript:
2.5 Imaging techniques for the assessment of liver status (Page 4 line 195-203)
The whole liver evaluation was performed under fasting conditions at the University of Navarra Clinic.
Liver steatosis was determined by ultrasonography (Siemens ACUSON S2000 and S3000, Erlangen,
Germany) in accordance with previously described methodology 47. The clinical classification was
established according to a 4-point scale: less than 5% (grade 0), 5-33% (grade 1), 33-66% (grade 2), and
greater than 66% (grade 3), as described elsewhere 48. Finally, Magnetic Resonance Imaging (Siemens
Aera 1,5 T) was used to determine the hepatic volume and the fat content of the liver (Dixon technique)
as reported by the manufacturer 32.
Indeed, the validated metabolomic test (OWLiver®-test) allowed to stratify the participants in normal
liver, steatosis or NASH.
1. Kupčová V, Fedelešová M, Bulas J, Kozmonová P, Turecký L. Overview of the Pathogenesis, Genetic, and Non-Invasive Clinical,
Biochemical, and Scoring Methods in the Assessment of NAFLD. Int J Environ Res Public Health. 2019 Sep 24;16(19):3570.
https://doi: 10.3390/ijerph16193570
2. Cantero I, Elorz M, Abete I, Marin BA, Herrero JI, Monreal JI, Benito A, Quiroga J, Martínez A, Huarte MP.; et al.
Ultrasound/elastography techniques, lipidomic and blood markers compared to magnetic resonance imaging in non-alcoholic
fatty liver disease adults. Int J Med Sci. 2019; 16, 75–83.
4. Would it be possible for the authors to clarify the sentence "After 6 months a total of 76
participants completed the evaluation."? Do they mean the dietary intervention that is
described later?
Thank you for your comments. We have tried to clarify this sentence in the manuscript as follows:
2.3. Dietary and lifestyle intervention (Page 4 line 165-168)
After 6 months of nutritional treatment, both AHA and FLiO groups (n= 34 and 36, respectively) achieved
comparable results in the evaluated main variables and no significant differences in the changes between
the intervention groups were found33. Therefore, participants were merged and compared together.
In any case, statistical adjustment by dietary group were performed when appropriate. Moreover, thanks
to the reviewer comments, we have also corrected some information concerning this issue as follows:
2.2 Study participants (Page 3, line 141-145)
A total of 98 men and women with overweight or obesity (Body Mass Index (BMI) ≥ 27.5 kg/m2 to < 40
kg/m2) between 40-80 years old and with hepatic steatosis confirmed by abdominal ultrasonography
fulfilled the selection criteria and were enrolled in the study 35. After 6 months a total of 70 participants
completed the evaluation.
5. It would be helpful if the authors could mention the specific time points they performed their
assessments in the material and methods section.
Thank you for your comments. Authors have tried to clarify this issue in the manuscript.
2.3. Dietary and lifestyle intervention (Page 4 line 160-168)
Two energy-restricted diets, AHA (n=41) and FLiO (n=45) were prescribed 33. Both diets applied an energy
restriction of 30% of the total energy requirements of each participant in order to achieve a loss of at least
3-5% of the initial body weight, in accordance with the recommendations of the American Association for
the Study of Liver Diseases guidelines (AASLD) 37. After 6 months of nutritional treatment, both AHA and
FLiO groups (n= 34 and 36, respectively) achieved comparable results in the evaluated main variables and
no significant differences in the changes between the intervention groups were found33. Therefore,
participants were merged and compared together.
Indeed, following reviewer suggestion a flow chart of study design has been also incorporated in the Page
6, line 280-315.
Figure 1. Design and flow-chart of participants in the Fatty Liver in Obesity (FLiO) study. AHA, American
Heart Association; FLI, Fatty Liver Index; FLiO, Fatty Liver in Obesity; GRS, Genetic Risk Score; GRSFLI,
Genetic Risk Score for FLI; GRSMRI, Genetic Risk Score for Magnetic Resonance Imaging; GRSOWL, Genetic
Risk Score for OWLiver®-test; MRI, Magnetic Resonance Imaging; SNPs, Single Nucleotide Polymorphisms.
6. It would be useful if the authors could provide a brief description of the quality control steps
they performed in their genetic analysis. Moreover, did they control for population
stratification?
Thank you for this question. As suggested by the reviewer, a brief description of the quality control steps
performed in the genetic analysis has been now included in the manuscript as follows:
2.7 SNP Selection and Genotyping (Page 5, line 226-240)
A total of 86 oral epithelium samples were collected with a liquid-based kit (ORAcollect-DNA, OCR-100,
DNA genotek Inc, Ottawa, Canada). Genomic DNA was isolated using the Maxwell® 16 Buccal Swab LEV
DNA Purification Kit (Promega Corp, Madison, WI, USA). The quality characterization was carried out by
dsDNA quantification (Qubit, ThermoFisher, Waltham, MA, USA). A pre-designed panel of 95 genetic
variants related to obesity and weight loss was applied and analyzed 10,50–52. More information about these
Collection of oral
epithelium samples
Nutritional intervention period
FLiO group
n=36
AHA group
n=34 6 months follow-up
FLiO group
n=45
Genotyped
participants
n=86
AHA group Baseline
n=41
Ultrasonography
proven liver steatosis
n=98
Pre-designed
panel of 95 SNPs
47 SNPs associated with the
improvement of hepatic health
23 SNPs with FLI (GRSFLI)
16 SNPs with OWLiver®-test (GRSOWL)
8 SNPs with MRI (GRSMRI)
Volunteers genotyped
after 6-months of
nutritional intervention
n=70
Calculation
of the GRS
Application
Randomization
obesity-related SNPs can be found in a previous report 28. Genotyping was performed by targeted next
generation sequencing on Ion Torrent PGM equipment (Thermo Fisher Scientific Inc., Waltham, MA,
USA)53, as published 50,54. Overall, the amplicon mean size was 185 bp. As quality control of the sequencing
process, the number of readings per amplifier and per sample were doubly checked, to make sure there
are more than 50x (in fact, it is above about 400x even the worst). Library construction was carried out
using a custom-designed panel and the Ion 198 AmpliSeq Library Kit 2.0 (Thermo Fisher Scientific) as per
the manufacturer’s protocol.
7. The authors mention that "The normality of the distribution was checked through Shapiro-
Wilk and Kolmogorov–Smirnov test." Can they clarify this point? Did they require both tests to
confirm normal distribution?
Thank you for your comments. As the reviewer comments, we applied both test for the assessment of
normality. If one of the two tests showed that variable was not normally distributed, non-parametric
statistics was used in its analysis [1,2]. Recent studies have suggested the Shapiro-Wilk test is a
powerful test for the assessment of normality [3]. However, it has some disadvantages showing
some difficulties with samples with many identical values [4] Therefore, we used the Kolmogorov-
Smirnov test as a complementary method for the assessment of normality, as well as histograms
and boxplots to visualize the distribution of the data [5].
1. Gilson M, Tauste Campo A, Chen X, Thiele A, Deco G. Nonparametric test for connectivity detection in multivariate
autoregressive networks and application to multiunit activity data. Netw Neurosci. 2017 Dec 1;1(4):357-380.
https://doi/10.1162/NETN_a_00019.
2. Kitchen CM. Nonparametric vs parametric tests of location in biomedical research. Am J Ophthalmol. 2009 Apr;147(4):571-2.
https://doi/10.1016/j.ajo.2008.06.031. PMID: 19327444; PMCID: PMC2743502.
3. Razali, N. M., & Wah, Y. B. (2011). Power comparisons of Shapiro-Wilk, Kolmogorov-Smirnov, Lilliefors and Anderson-Darling
tests. Journal of Statistical Modeling and Analytics, 2(1), 21–33.
4. AJ, Li J, Conley C. Informal versus formal judgment of statistical models: The case of normality assumptions. Psychon Bull Rev.
2021 Mar 3. doi: 10.3758/s13423-021-01879-z.
5. DF Statistics for Goodness of Fit and Some Comparisons. Journal of the American Statistical Association. 69 (347): 730–
737. doi:10.2307/2286009. JSTOR 2286009.
8. Did the authors consider correcting for multiple tests?
Thank you for this question. The study proposed in this manuscript was designed as a hypothesis-driven
analyses research which overcome the need for correction of multiple comparisons. Moreover, three
different Genetic Risk Score (GRS) were derived in order to evaluate the cumulative effect of all the SNPs
that showed an association with the improvement of hepatic health, measured by three different noninvasive
methods (FLI, Magnetic Resonance Imaging and OWLiver®-test), after 6-month nutritional
intervention, which specifically assessed liver function, fat content and lipid metabolism, respectively.
Therefore, we selected those SNPs showing a conventionally accepted of at least a marginal statistical
trend (p<0.10) and excluding those with low sample (<10%) in either category or due to collinearity.
Genotypes with similar effects were clustered in a single category. Then, in order to estimate the
proportion of trait variability explained by the GRS, we performed a linear regression analysis, where the
major contributors of the % of change in Fatty Liver Index, change in OWLiver®-test and change in liver
fat content (MRI) were the GRSFLI (p<0.001), GRSOWL (p=0.011) and GRSMRI (p<0.001), respectively. It is also
important to mention that this approach has been also used in previous studies [2-4].
1. Merino, J., Leong, A., Posner, D. C., Porneala, B., Masana, L., Dupuis, J., & Florez, J. C. (2017). Genetically Driven Hyperglycemia
Increases Risk of Coronary Artery Disease Separately from Type 2 Diabetes. Diabetes care, 40(5), 687–693.
https://doi.org/10.2337/dc16-2625
2. Igo RP Jr, Kinzy TG, Cooke Bailey JN. Genetic Risk Scores. Curr Protoc Hum Genet. 2019 Dec;104(1):e95.https://doi.org/
10.1002/cphg.95
3. Ramos-Lopez O, Riezu-Boj JI, Milagro FI, Cuervo M, Goni L, Martinez JA. Interplay of an Obesity-Based Genetic Risk Score with
Dietary and Endocrine Factors on Insulin Resistance. Nutrients. 2019 Dec 21;12(1):33. https://doi/10.3390/nu12010033. PMID:
31877696; PMCID: PMC7019905.
4. Sun D, Zhou T, Li X, Heianza Y, Liang Z, Bray GA, Sacks FM, Qi L. Genetic Susceptibility, Dietary Protein Intake, and Changes of
Blood Pressure: The POUNDS Lost Trial. Hypertension. 2019 Dec;74(6):1460-1467.
https://doi/10.1161/HYPERTENSIONAHA.119.13510. Epub 2019 Oct 28
9. Did the authors adjust their results based on which nutritional group the participants belong
to?
Thank you for this interesting question. In addition to genetic variants, other conventional factors of
personalization were analyzed, including age, sex, and the following variables at baseline: insulin (U/mL),
FGF-21 (pg/ml) and protein (%), as well as potential interaction introducing the corresponding interaction
terms in the models. In this sense, other potential modifiers were previously tested, in particular the
dietary group the participants belong to, even though it didn´t improved the model and no significant
differences were found. Moreover, we have added this information in the discussion text as follows:
4. Discussion (Page 14, line 573-581)
Moreover, for the purpose of explaining the variability on the improvement in hepatic functionality (FLI),
liver fat content (by MRI) and lipidomic (OWLiver®-test), linear regression models were performed. The
predictive accuracy of all models substantially improved when combining each of the previously
mentioned SNPs in the multiple linear regression models, which is in line with previous studies70. In order
to ameliorate these results, each regression model was fitted by sex, age and NAFLD related variables
such as inflammatory biomarkers or dietary compounds. Other variables such as the nutritional group of
the participants was also considered, even though no significant differences were found.
10. It would be helpful if the authors could mention in table 1 how many of the participants were
included in each nutritional group.
Thank you for your comments. We have revised and modified this issue throughout the manuscript as
follows:
2.3. Dietary and lifestyle intervention (Page 4 line 160-168)
Two energy-restricted diets, AHA (n=41) and FLiO (n=45) were prescribed 33. Both diets applied an energy
restriction of 30% of the total energy requirements of each participant in order to achieve a loss of at least
3-5% of the initial body weight, in accordance with the recommendations of the American Association for
the Study of Liver Diseases guidelines (AASLD) 37. After 6 months of nutritional treatment, both AHA and
FLiO groups (n= 34 and 36, respectively) achieved comparable results in the evaluated main variables and
no significant differences in the changes between the intervention groups were found 33. Therefore,
participants were merged and compared together.
33. Marin-Alejandre, B. A.; Abete, I.; Cantero, I.; Monreal, J. I.; Martinez-echeverria, A.; Uriz-otano, J. I. The Metabolic and Hepatic
Impact of Two Personalized Dietary Strategies in Subjects with Obesity and Nonalcoholic Fatty Liver Disease: The Fatty Liver in
Obesity ( FLiO ) Randomized Controlled Trial. Nutrients 2019, 11 (10). https://doi.org/10.3390/nu11102543.
Indeed, a flow-chart of study design has been also incorporated in the Page 6, line 280-315

Reviewer 2 Report

In this paper entitled “Three different genetic risk scores based on Fatty Liver Index, Magnetic Resonance Imaging and lipidomic for a nutrigenetic personalized management of NAFLD: the Fatty Liver in Obesity study” Nuria Perez-Diaz-del-Campo et al. tried to evaluate if the genetic background linked to NAFLD-related factors may influence hepatic amelioration after diet. Authors, evaluating 95 patients with NAFLD before and after 6-months weight-loss nutritional treatment concluded  that “three different genetic scores can be useful for the personalized management of NAFLD, whose treatment must rely on specific dietary recommendations guided by the measurement of specific genetic biomarkers”.

Even if the  topic of the paper is of great interest, the authors’ conclusions are not supported by the data included in the paper. The sample size was not estimated with the aim to evaluate the genetic risk factors for these patients, and I don’t think we can accept these conclusions only after the evaluation of 95 subjects.

Author Response

We would like to thank the reviewer for their thoughtful comments and suggestions for improvement.
We believe that in addressing these comments, this revised manuscript is considerably ameliorated.
Changes in the manuscript have been marked up using the “Track Changes” and are shown in this
document as well. A detailed point-by-point response to the reviewer is provided after each comment.
RESPONSE TO REVIEWER #2 COMMENTS
In this paper entitled “Three different genetic risk scores based on Fatty Liver Index, Magnetic Resonance
Imaging and lipidomic for a nutrigenetic personalized management of NAFLD: the Fatty Liver in Obesity
study” Nuria Perez-Diaz-del-Campo et al. tried to evaluate if the genetic background linked to NAFLDrelated
factors may influence hepatic amelioration after diet. Authors, evaluating 95 patients with NAFLD
before and after 6-months weight-loss nutritional treatment concluded that “three different genetic
scores can be useful for the personalized management of NAFLD, whose treatment must rely on specific
dietary recommendations guided by the measurement of specific genetic biomarkers”.
Even if the topic of the paper is of great interest, the authors’ conclusions are not supported by the data
included in the paper. The sample size was not estimated with the aim to evaluate the genetic risk factors
for these patients, and I don’t think we can accept these conclusions only after the evaluation of 95
subjects.
We would like to thank the reviewer for these constructive comments. The sample size was calculated
with weight loss as the main outcome, based on the current recommendations of the AASLD to
ameliorate NAFLD features [Reference 24 in the manuscript]. In this sense, and according to previous
studies [Reference 33 in the manuscript], the sample size was calculated to detect a difference of 1.0 (1.5
kg) between both dietary groups (AHA vs. FLiO) in their reduction of weight, with a 95% confidence
interval (α = 0.05) and a statistical power of 80% (β = 0.8). This approach estimated a total of 36
participants per study group but considering the estimated dropout rate of 20–30% (according to the
experience of the research group), 50 subjects were included in each arm of the study. However, two
subjects were excluded from the AHA group due to important alterations in the initial assessment of
biochemical parameters, which required medical treatment. This trial started with 98 subjects but only
86 epithelium buccal cells of them were collected. Moreover, after six months of weight loss intervention,
a total of 70 participants had complete information and epithelium buccal cells to carry out the objective
of this study. Despite that, the outcomes were scientifically adequate and expectable.
24. Rinella ME, Tacke F, Sanyal AJ, Anstee QM (2019) Report on the AASLD/EASL joint workshop on clinical trial endpoints in
NAFLD. J Hepatol. 71:823–833. https://doi.org/10.1016/j. jhep.2019.04.019
33. Marin-Alejandre BA, Abete I, Cantero I, Monreal JI, Elorz M, Herrero JI, Benito-Boillos A, Quiroga J, Martinez-Echeverria A,
Uriz-Otano JI et al (2019) The metabolic and hepatic impact of two personalized dietary strategies in subjects with obesity
and nonalcoholic fatty liver disease: the fatty liver in obesity (FLiO) randomized controlled trial. Nutrients. 11:2543. https
://doi.org/10.3390/nu11102543
As reviewer suggests, sample size is a limitation of this article and although type I and type II errors
cannot be discarded, results reported in the manuscript are plausible and followed the hypothesized
outcomes, which give support to our findings. In addition, we performed a genetic risk score to reduce
the number of variables (95 SNPs) in order to detect major differences and associations with the
improvement of hepatic health after the 6-months nutritional intervention. In this sense, recent research
with similar study design have also stablished different relationships between genetic risk scores and
obesity, insulin sensitivity subjects [1,2] or different metabolic traits [3]. Indeed, other studies with a
similar design and sample size have also reported interesting associations using this statistical approach
[4-7]. Furthermore, this research is a proof of principle and because of that, future validations in different
populations and bigger cohorts may be needed.
1. Franck M, de Toro-Martín J, Guénard F, Rudkowska I, Lemieux S, Lamarche B, Couture P, Vohl MC. Prevention of Potential
Adverse Metabolic Effects of a Supplementation with Omega-3 Fatty Acids Using a Genetic Score Approach. Lifestyle Genom.
2020;13(1):32-42. https://doi.org/10.1159/000504022
2. Goni L, Cuervo M, Milagro FI, Martínez JA. A genetic risk tool for obesity predisposition assessment and personalized nutrition
implementation based on macronutrient intake. Genes Nutr. 2015 Jan;10(1):445. https://doi.org/10.1007/s12263-014-0445-
z.
3. Surendran, S., Aji, A. S., Ariyasra, U., Sari, S. R., Malik, S. G., Tasrif, N., Yani, F. F., Lovegrove, J. A., Sudji, I. R., Lipoeto, N. I., &
Vimaleswaran, K. S. (2019). A nutrigenetic approach for investigating the relationship between vitamin B12 status and
metabolic traits in Indonesian women. Journal of diabetes and metabolic disorders, 18(2), 389–399.
https://doi.org/10.1007/s40200-019-00424-z
4. Klimentidis YC, Bea JW, Lohman T, Hsieh PS, Going S, Chen Z. High genetic risk individuals benefit less from resistance exercise
intervention. Int J Obes (Lond). 2015 Sep;39(9):1371-5. doi: 10.1038/ijo.2015.78. Epub 2015 Apr 30.
5. León-Mimila P, Vega-Badillo J, Gutiérrez-Vidal R, Villamil-Ramírez H, Villareal-Molina T, Larrieta-Carrasco E, López-Contreras
BE, Kauffer LR, Maldonado-Pintado DG, Méndez-Sánchez N, Tovar AR, Hernández-Pando R, Velázquez-Cruz R, Campos-Pérez
F, Aguilar-Salinas CA, Canizales-Quinteros S. A genetic risk score is associated with hepatic triglyceride content and nonalcoholic
steatohepatitis in Mexicans with morbid obesity. Exp Mol Pathol. 2015 Apr;98(2):178-83. doi:
10.1016/j.yexmp.2015.01.012.
6. Jeppesen PB, Gilroy R, Pertkiewicz M, Allard JP, Messing B, O'Keefe SJ. Randomised placebo-controlled trial of teduglutide in
reducing parenteral nutrition and/or intravenous fluid requirements in patients with short bowel syndrome. Gut. 2011
Jul;60(7):902-14. doi: 10.1136/gut.2010.218271.
7. Rudkowska I, Guénard F, Julien P, Couture P, Lemieux S, Barbier O, Calder PC, Minihane AM, Vohl MC. Genome-wide association
study of the plasma triglyceride response to an n-3 polyunsaturated fatty acid supplementation. J Lipid Res. 2014
Jul;55(7):1245-53. doi: 10.1194/jlr.M045898.
4. Discussion (Page 14-15, line 617-632)
As for drawbacks of this research: Firstly, liver biopsy results were not available to corroborate the precise
diagnosis of patients 57. Nonetheless, in this research we carried out a complete evaluation of the liver
status by means of validated and widely used techniques as well as blood biomarkers and hepatic indexes,
which are affordable and practical methods to use in health assessment. Second, the sample size and the
enrollment of subjects are not very large. For this reason, these models may be further validated in
different populations, to establish whether they might represent a reliable and accurate, “non-invasive
alternative” to liver biopsy. Also, the role of new SNPs associated with excessive adiposity and
accompanying metabolic alterations through a GRS approach needs to be re-explored, but our
contribution re-state the value of the genetic make-up when prescribing personalized diets. Thirdly, type
I and type II errors cannot be completely ruled out, especially those related to the selection of SNPs to be
introduced into the GRS. However, due to the use of less stringent P value thresholds compared to
association studies of single variants, genomic profile risk scoring analyses can tolerate, at balance, some
of these biases, as previously reviewed 87.

Reviewer 3 Report

The study evaluated if genetic risks were associated with non-invasive markers of hepatic health. Then used these markers to investigate if dietary/physical interventions were effective in improving hepatic health in carriers of genetic variants. They used a known set of 95 SNPs/variants associated with obesity/inflammation and tested their association with 3 measures of hepatic health in a small cohort of 86. A combination of GRS_clin markers improved prediction. The GRS+clin markers were then used to investigate whether the interventions in this cohort reduced obesity. They found specific gene variants associated with the three clinical tests and that specific intervention to reduce obesity can be targeted in individuals carrying a certain set of risk alleles/variants.

I have two major queries.
1. If I understand correctly, the authors chose 95 SNPs associated with obesity, then used a subset of these that were associated with improving hepatic health in their cohort. This information was then used to test improved hepatic health on 6-month intervention in this cohort. This seems a circular argument. The validation should have been performed in independent set of cohorts, not the population they were derived from. If I have misunderstood the study principal, then perhaps it was not explained clearly enough and needs to be addressed. Perhaps a flow diagram of study design will be helpful in understanding the steps better.
2. The 2nd concern is the use of a large number (~95) of SNPs as variables in a small cohort size (~86), which is not ideal statistically. The p-val threshold is high.

Other 
3. When hepatic health is a main measure of the study it is important to include genetic variants that predispose some to liver injury. Why were SNPs known to be associated with hepatic injury in NAFLD (PNPLA3, TM6SF2, HSD17B13 etc) not included to see the vulnerability in carriers of these risk alleles in addition to obesity related SNPs.
4. There are 97 references which are too many for original article of this length

5. Table 1. Seems the numbers are swapped for adiponectin and leptin in <50 y and >50 y

6. Line 314: How was the risk calculated at <9 for GRS-FLi with 17 SNPs? Was minor allele taken into consideration?

Author Response

We would like to thank the reviewer for their thoughtful comments and suggestions for improvement.
We believe that in addressing these comments, this revised manuscript is considerably ameliorated.
Changes in the manuscript have been marked up using the “Track Changes” and are shown in this
document as well. A detailed point-by-point response to the reviewer is provided after each comment.
RESPONSE TO REVIEWER #3 COMMENTS
The study evaluated if genetic risks were associated with non-invasive markers of hepatic health. Then
used these markers to investigate if dietary/physical interventions were effective in improving hepatic
health in carriers of genetic variants. They used a known set of 95 SNPs/variants associated with
obesity/inflammation and tested their association with 3 measures of hepatic health in a small cohort of
86. A combination of GRS_clin markers improved prediction. The GRS+clin markers were then used to
investigate whether the interventions in this cohort reduced obesity. They found specific gene variants
associated with the three clinical tests and that specific intervention to reduce obesity can be targeted in
individuals carrying a certain set of risk alleles/variants.
I have two major queries.
1. If I understand correctly, the authors chose 95 SNPs associated with obesity, then used a subset of
these that were associated with improving hepatic health in their cohort. This information was then
used to test improved hepatic health on 6-month intervention in this cohort. This seems a circular
argument. The validation should have been performed in independent set of cohorts, not the
population they were derived from. If I have misunderstood the study principal, then perhaps it was
not explained clearly enough and needs to be addressed. Perhaps a flow diagram of study design will
be helpful in understanding the steps better.
Thank you for your comments. We have used a pre-designed panel of 95 SNPs associated with obesity.
Once the 95 SNPs were genotyped, three individual GRS were calculated for the change of each noninvasive
diagnostic method (FLI, MRI and OWLiver®-test) to test the improvement of hepatic health after
6-month nutritional intervention in this cohort. Thus, we obtained 47 SNPs associated with FLI, OWLiver®-
test and MRI (See Supplementary material Figure S1). The analysis should be considered as a proof-ofprinciple
showing that some specific genes alleles regulating liver functions, fat content and lipid
metabolism are differentially involved and are important to prescribe diets and manage liver diseases.
Supplementary material
Figure S1. Venn diagram showing the number of SNPs associated with each NAFLD non-invasive
diagnostic methods. GRS, Genetic Risk Score; MRI, Magnetic Resonance Imaging; FLI, Fatty Liver Index;
OWL, OWLiver®-test.
Indeed, the main objective of this research was not to perform a validation since it is a proof-of concept,
where the purpose was to evaluate if the genetic background linked to NAFLD-related factors such as
obesity may influence hepatic amelioration following the impact of the genetic make-up based on GRS.
Moreover, in agreement with the reviewer, we have tried to clarify this issue throughout the manuscript:
2.7. SNP Selection and Genotyping (Page 5 line 231-233)
A pre-designed panel of 95 genetic variants related to obesity and weight loss was applied and
analyzed10,50–52. More information about these obesity-related SNPs can be found in a previous report 28.
10. Goni L, Qi L, Cuervo M, Milagro FI, Saris WH, MacDonald IA, Langin D, Astrup A, Arner P, Oppert JM, Svendstrup M, Blaak EE,
Sørensen TI, Hansen T, Martínez JA. Effect of the interaction between diet composition and the PPM1K genetic variant on insulin
resistance and β cell function markers during weight loss: results from the Nutrient Gene Interactions in Human Obesity: implications
for dietary guidelines (NUGENOB) randomized trial. Am J Clin Nutr. 2017 Sep;106(3):902-908.
https://doi.org/10.3945/ajcn.117.156281.
51. Ramos-Lopez, O.; Milagro, F. I.; Allayee, H.; Chmurzynska, A.; Choi, M. S.; Curi, R.; De Caterina, R.; Ferguson, L. R.; Goni, L.; Kang,
J. X.; et al. Guide for Current Nutrigenetic, Nutrigenomic, and Nutriepigenetic Approaches for Precision Nutrition Involving the
3 elements included
exclusively in GRSMRI:
rs6861681 (CPEB4)
rs1440581 (PPM1K)
rs1799883 (FABP2)
10 elements included
exclusively in GRSOWL:
rs1175544 (PPARG)
rs1797912 (PPARG)
rs1386835 (PPARG)
rs709158 (PPARG)
rs1175540 (PPARG)
rs1801260 (CLOCK)
rs12502572 (UCP1)
rs8179183 (LEPR)
rs894160 (PLIN1)
rs4731426 (LEP)
17 elements included
exclusively in "GRSFLI":
rs1801133 (MTHFR)
rs1055144 (NFE2L3)
rs17817449 (FTO)
rs8050136 (FTO)
rs3751812 (FTO)
rs9939609 (FTO)
rs2075577 (UCP3)
rs324420 (FAAH)
rs1121980 (FTO)
rs2419621 (ACSL5)
rs1558902 (FTO)
rs3123554 (CNR2FUCA1)
rs6567160 (MC4R)
rs660339 (UCP2)
rs2605100 (LYPLAL1)
rs1800629 (TNFAPROMOTOR)
rs4994 (ADRB3)
3 common elements in
GRSFLI and GRSOWL
rs6123837 (GNAS)
rs571312 (MC4R)
rs6536991 (UCP1)
2 common elements in
GRSFLI and GRSMRI:
rs494874 (ABCB11)
rs7359397_(SH2B1)
2 common elements in
GRSOWL and GRSMRI:
rs7498665 (SH2B1)
rs4929949 (STK33)
1 common element in GRSFLI,
GRSOWL and GRSMRI:
rs2959272 (PPARG)
GRSOWL
GRSMRI
GRSFLI
Prevention and Management of Chronic Diseases Associated with Obesity. J. Nutrigenet. Nutrigenomics 2017, 10 (1–2), 43–62.
https://doi.org/10.1159/000477729.
52. Heianza, Y.; Ma, W.; Huang, T.; Wang, T.; Zheng, Y.; Smith, S. R.; Bray, G. A.; Sacks, F. M.; Qi, L. Macronutrient Intake-Associated
FGF21 Genotype Modifies Effects of Weight-Loss Diets on 2-Year Changes of Central Adiposity and Body Composition: The POUNDS
Lost Trial. Diabetes Care 2016, 39 (11), 1909–1914. https://doi.org/10.2337/dc16-1111.
53. Goni, L.; Cuervo, M.; Milagro, F. I.; Martínez, J. A. Gene-Gene Interplay and Gene-Diet Interactions Involving the MTNR1B
Rs10830963 Variant with Body Weight Loss. J. Nutrigenet. Nutrigenomics 2015, 7, 232–242. https://doi.org/10.1159/000380951.
28. Ramos-Lopez, O.; Cuervo, M.; Goni, L.; Milagro, F. I.; Riezu-Boj, J. I.; Martinez, J. A. Modeling of an Integrative Prototype Based
on Genetic, Phenotypic, and Environmental Information for Personalized Prescription of Energy-Restricted Diets in
Overweight/Obese Subjects. Am. J. Clin. Nutr. 2020, 111 (2), 459–470. https://doi.org/10.1093/ajcn/nqz286.
Abstract (Page 1 line 40-41)
Also, a pre-designed panel of 95 genetic variants related to obesity and weight loss was applied and
analyzed.
Furthermore, a flow chart of study design has been done as suggested: (Page 6 line 280-315)
Figure 1. Design and flow chart of participants in the Fatty Liver in Obesity (FLiO) study. FLI, Fatty Liver
Index; GRS, Genetic Risk Score; GRSFLI, Genetic Risk Score for FLI; GRSMRI, Genetic Risk Score for Magnetic
Resonance Imaging; GRSOWL, Genetic Risk Score for OWLiver®-test; MRI, Magnetic Resonance Imaging;
SNPs, Single Nucleotide Polymorphisms.
Collection of oral
epithelium samples
Nutritional intervention period
FLiO group
n=36
AHA group
n=34 6 months follow-up
FLiO group
n=45
Genotyped
participants
n=86
AHA group Baseline
n=41
Ultrasonography
proven liver steatosis
n=98
Pre-designed
panel of 95 SNPs
47 SNPs associated with the
improvement of hepatic health
23 SNPs with FLI (GRSFLI)
16 SNPs with OWLiver®-test (GRSOWL)
8 SNPs with MRI (GRSMRI)
Volunteers genotyped
after 6-months of
nutritional intervention
n=70
Calculation
of the GRS
Application
Randomization
2. The 2nd concern is the use of a large number (~95) of SNPs as variables in a small cohort size (~86),
which is not ideal statistically. The p-val threshold is high.
Thank you for your comments. The reviewer is right, the relatively small sample size and the absence of a
reference group are important limitations of this study. However, this is a screening study, which was
designed to search possibilities of different GRS to seek precision nutrition. Therefore, the p-value is high,
but it is accepted in scientific studies. [Reference 28 in the manuscript].
28. Ramos-Lopez, O.; Cuervo, M.; Goni, L.; Milagro, F. I.; Riezu-Boj, J. I.; Martinez, J. A. Modeling of an Integrative Prototype Based
on Genetic, Phenotypic, and Environmental Information for Personalized Prescription of Energy-Restricted Diets in
Overweight/Obese Subjects. Am. J. Clin. Nutr. 2020, 111 (2), 459–470. https://doi.org/10.1093/ajcn/nqz286.
Moreover, this limitation has been recognized in the discussion of the text as follows:
4. Discussion (Page 14-15, line 617-632)
As for drawbacks of this research: Firstly, liver biopsy results were not available to corroborate the precise
diagnosis of patients 57. Nonetheless, in this research we carried out a complete evaluation of the liver
status by means of validated and widely used techniques as well as blood biomarkers and hepatic indexes,
which are affordable and practical methods to use in health assessment. Second, the sample size and the
enrollment of subjects are not very large. For this reason, these models may be further validated in
different populations, to establish whether they might represent a reliable and accurate, “non-invasive
alternative” to liver biopsy. Also, the role of new SNPs associated with excessive adiposity and
accompanying metabolic alterations through a GRS approach needs to be re-explored, but our
contribution re-state the value of the genetic make-up when prescribing personalized diets. Thirdly, type
I and type II errors cannot be completely ruled out, especially those related to the selection of SNPs to be
introduced into the GRS. However, due to the use of less stringent P value thresholds compared to
association studies of single variants, genomic profile risk scoring analyses can tolerate, at balance, some
of these biases, as previously reviewed 87.
Furthermore, even if type I and type II errors cannot be discarded because of the small cohort size, results
reported in the manuscript are plausible and followed the hypothesized outcomes which give support to
our findings. Also, this research is a proof of principle and because of that, future validations in different
populations and bigger cohorts is need. In this sense, recent research with similar study design have also
stablished different relationship between genetic risk scores and obesity, insulin sensitivity subjects [1,2]
or different metabolic traits [3]. Indeed, other studies with similar design and sample size have also
stablished interesting associations using this statistical approach [4-7]. Furthermore, this research is a
hypothesis driven study, and because of that, future validations in different populations and bigger
cohorts are convenient to envisage the role of genetics in liver management, but it is evident that liver
functionality, composition and metabolism are regulated by different sets of genes with value on
diagnosis.
1. Franck M, de Toro-Martín J, Guénard F, Rudkowska I, Lemieux S, Lamarche B, Couture P, Vohl MC. Prevention of Potential
Adverse Metabolic Effects of a Supplementation with Omega-3 Fatty Acids Using a Genetic Score Approach. Lifestyle Genom.
2020;13(1):32-42. https://doi.org/10.1159/000504022
2. Goni L, Cuervo M, Milagro FI, Martínez JA. A genetic risk tool for obesity predisposition assessment and personalized nutrition
implementation based on macronutrient intake. Genes Nutr. 2015 Jan;10(1):445. https://doi.org/10.1007/s12263-014-0445-
z.
3. Surendran, S., Aji, A. S., Ariyasra, U., Sari, S. R., Malik, S. G., Tasrif, N., Yani, F. F., Lovegrove, J. A., Sudji, I. R., Lipoeto, N. I., &
Vimaleswaran, K. S. (2019). A nutrigenetic approach for investigating the relationship between vitamin B12 status and
metabolic traits in Indonesian women. Journal of diabetes and metabolic disorders, 18(2), 389–399.
https://doi.org/10.1007/s40200-019-00424-z
4. Klimentidis YC, Bea JW, Lohman T, Hsieh PS, Going S, Chen Z. High genetic risk individuals benefit less from resistance exercise
intervention. Int J Obes (Lond). 2015 Sep;39(9):1371-5. doi: 10.1038/ijo.2015.78. Epub 2015 Apr 30.
5. León-Mimila P, Vega-Badillo J, Gutiérrez-Vidal R, Villamil-Ramírez H, Villareal-Molina T, Larrieta-Carrasco E, López-Contreras
BE, Kauffer LR, Maldonado-Pintado DG, Méndez-Sánchez N, Tovar AR, Hernández-Pando R, Velázquez-Cruz R, Campos-Pérez
F, Aguilar-Salinas CA, Canizales-Quinteros S. A genetic risk score is associated with hepatic triglyceride content and nonalcoholic
steatohepatitis in Mexicans with morbid obesity. Exp Mol Pathol. 2015 Apr;98(2):178-83. doi:
10.1016/j.yexmp.2015.01.012.
6. Jeppesen PB, Gilroy R, Pertkiewicz M, Allard JP, Messing B, O'Keefe SJ. Randomised placebo-controlled trial of teduglutide in
reducing parenteral nutrition and/or intravenous fluid requirements in patients with short bowel syndrome. Gut. 2011
Jul;60(7):902-14. doi: 10.1136/gut.2010.218271.
7. Rudkowska I, Guénard F, Julien P, Couture P, Lemieux S, Barbier O, Calder PC, Minihane AM, Vohl MC. Genome-wide association
study of the plasma triglyceride response to an n-3 polyunsaturated fatty acid supplementation. J Lipid Res. 2014
Jul;55(7):1245-53. doi: 10.1194/jlr.M045898.
3. When hepatic health is a main measure of the study it is important to include genetic variants that
predispose some to liver injury. Why were SNPs known to be associated with hepatic injury in NAFLD
(PNPLA3, TM6SF2, HSD17B13 etc) not included to see the vulnerability in carriers of these risk alleles in
addition to obesity related SNPs.
Thank you for your comments. We agree with reviewer, genetic variants related to metabolism of
triglycerides (PNPLA3), hepatic lipid metabolism (TM6SF2) or involved in the regulation of de novo
lipogenesis (GCKR), have been specifically associated, among others, with NAFLD [1,2].
However, this experiment was designed as a proof-of concept, where authors performed a genetic risk
score using a pre-designed panel which included genetic variants related with obesity and weight-loss. In
this sense, the current guidance statement of the American Association for the Study of Liver Diseases
(AASLD), has highlighted that a weight-loss of at least 3-5% of body weight seems necessary to improve
hepatic steatosis, but a greater weight loss (7-10%) is needed to improve histopathological characteristics
of NASH, including fibrosis [3]. Moreover, available scientific bibliography [4-6], underline that for a better
understanding of the pathogenesis of the disease, less prevalent genetic variants (MAF <5%), as well as
different metabolic pathways may also be a key factor to be accounted for. Interestingly, genes regulating
liver physiopathological mechanisms were ascertained with a role on diagnosis and management of
NAFLD.
1. Choudhary, Narendra Singh, and Ajay Duseja. “Genetic and epigenetic disease modifiers: non-alcoholic fatty liver disease (NAFLD)
and alcoholic liver disease (ALD).” Translational gastroenterology and hepatology vol. 6 2. 5 Jan. 2021, doi:10.21037/tgh.2019.09.06
2. Eslam M, Valenti L, Romeo S. Genetics and epigenetics of NAFLD and NASH: Clinical impact. J Hepatol. 2018 Feb;68(2):268-279.
doi: 10.1016/j.jhep.2017.09.003.
3. Chalasani, N., Younossi, Z., Lavine, J. E., Charlton, M., Cusi, K., Rinella, M., Harrison, S. A., Brunt, E. M., & Sanyal, A. J. (2018). The
diagnosis and management of nonalcoholic fatty liver disease: Practice guidance from the American Association for the Study of
Liver Diseases. Hepatology, 67(1), 328–357.
4. Pelusi, S., Baselli, G., Pietrelli, A. et al. Rare Pathogenic Variants Predispose to Hepatocellular Carcinoma in Nonalcoholic Fatty
Liver Disease. Sci Rep 9, 3682 (2019). https://doi.org/10.1038/s41598-019-39998-2
5. Sookoian S, Pirola CJ. Genetic predisposition in nonalcoholic fatty liver disease. Clin Mol Hepatol. 2017;23(1):1-12.
doi:10.3350/cmh.2016.0109
6. Dongiovanni P, Anstee QM, Valenti L. Genetic predisposition in NAFLD and NASH: impact on severity of liver disease and response
to treatment. Curr Pharm Des. 2013;19(29):5219-5238. doi:10.2174/13816128113199990381
4. There are 97 references which are too many for original article of this length
Thank you for your comments. In agreement with the reviewer, we have revised and removed some of
the references.
5. Table 1. Seems the numbers are swapped for adiponectin and leptin in <50 y and >50 y
The reviewer is right. We had a technical error. We have corrected adiponectin and leptin concentrations
in <50 y and >50 y in Table 1.
6. Line 314: How was the risk calculated at <9 for GRS-FLi with 17 SNPs? Was minor allele taken into
consideration?
Thank you for this question. To study the genetic risk association with NAFLD evaluated by the Fatty Liver
Index, 23 obesity-related genetic variants were chosen from the pre-designed panel, because they were
statistically or marginally associated with the improvement in hepatic functionality measured by FLI (See
supplementary material Figure S1). Of those, 17 SNPs were exclusively related to FLI: rs1801133 (MTHFR),
rs1055144 (NFE2L3), rs17817449 (FTO), rs8050136 (FTO), rs3751812 (FTO), rs9939609 (FTO), rs2075577
(UCP3), rs324420 (FAAH), rs1121980 (FTO), rs2419621 (ACSL5), rs1558902 (FTO), rs3123554
(CNR2/FUCA1), rs6567160 (MC4R), rs660339 (UCP2), rs2605100 (LYPLAL1), rs1800629
(TNFAPROMOTOR), rs4994 (ADRB3).
To evaluate the combined effects of the previously selected SNPs on the change of FLI, an individual GRS
was calculated by summing the number of risk alleles carried by each subject. The maximum value of the
genetic risk score among participants was 18. Therefore, the sample was stratified by the median, into
“low genetic risk group”, those with a GRSFLI ≤ 9, and into a “high genetic risk group,” those with a
GRSFLI > 9 risk alleles (these groups could include the minor allele frequency dominant allele or not).
Finally, we compared variables changes according to the median of GRSFLI < 9 and GRSFLI ³ 9.
This approach was also applied concerning the other GRSOWL and GRSMRI with a diagnostic role

Round 2

Reviewer 1 Report

Manuscript ID: diagnostics-1222244

Title: "Three different genetic risk scores based on Fatty Liver Index, Magnetic Resonance Imaging and lipidomic for a nutrigenetic personalized management of NAFLD: the Fatty Liver in Obesity study".

Authors: Nuria Perez-Diaz-del-Campo, et al.

The authors have satisfactorily responded to my comments and suggestions. They have made the necessary changes to the manuscript and improved the quality of their paper. The revised manuscript is an overall interesting and well-written paper with significant findings. There are no further considerations.

Reviewer 2 Report

the manuscript has not been improved sufficiently. to warrant publication in Diagnostics.

I suggest to change the title of the manuscript and the conclusions.

If the sample size has not been calculated to estimate the genetic risk factor, as you mentioned in the discussion, you can not conclude as in the abstract: 

These results demonstrate that three different genetic scores can be useful for the personalized management of NAFLD, whose treatment must rely on specific dietary recommendations guided by the measurement of specific genetic biomarkers.

This manuscript is a resubmission of an earlier submission. The following is a list of the peer review reports and author responses from that submission.